# Genetic variation, environment and demography intersect to shape Arabidopsis defense metabolite variation across Europe

Ella Katz[1], Jia-Jie Li[1], Benjamin Jaegle[2], Haim Ashkenazy[3], Shawn R Abrahams[4], Clement Bagaza[5], Samuel Holden[5], Chris J Pires[4], Ruthie Angelovici[5], Daniel J Kliebenstein[1,6]*

[1]Department of Plant Sciences, University of California, Davis, Davis, United States; [2]Gregor Mendel Institute, Austrian Academy of Sciences, Vienna Biocenter (VBC), Vienna, Austria; [3]Department of Molecular Biology, Max Planck Institute for Developmental Biology, Tübingen, Germany; [4]Division of Biological Sciences, Bond Life Sciences Center, University of Missouri, Columbia, United States; [5]Division of Biological Sciences, Interdisciplinary Plant Group, Christopher S. Bond Life Sciences Center, University of Missouri, Columbia, United States; [6]DynaMo Center of Excellence, University of Copenhagen, Frederiksberg, Denmark

*For correspondence:
kliebenstein@ucdavis.edu

**Abstract** Plants produce diverse metabolites to cope with the challenges presented by complex and ever-changing environments. These challenges drive the diversification of specialized metabolites within and between plant species. However, we are just beginning to understand how frequently new alleles arise controlling specialized metabolite diversity and how the geographic distribution of these alleles may be structured by ecological and demographic pressures. Here, we measure the variation in specialized metabolites across a population of 797 natural *Arabidopsis thaliana* accessions. We show that a combination of geography, environmental parameters, demography and different genetic processes all combine to influence the specific chemotypes and their distribution. This showed that causal loci in specialized metabolism contain frequent independently generated alleles with patterns suggesting potential within-species convergence. This provides a new perspective about the complexity of the selective forces and mechanisms that shape the generation and distribution of allelic variation that may influence local adaptation.

## Introduction

Continuous and dynamic change in a plant's habitat/environment creates a complex system to which a plant must adapt. Central to this adaptation are the production and accumulation of different metabolites ranging from signaling hormones, primary metabolites to a wide array of multi-functional specialized metabolites (*Erb and Kliebenstein, 2020*; *Hanower and Brzozowska, 1975*; *Hayat et al., 2012*; *Kim et al., 2012*; *Kliebenstein, 2004*; *Malcolm, 1994*; *Thakur and Rai, 1982*; *Wolters and Jürgens, 2009*; *Yang et al., 2000*). The complete blend, chemotype, of these metabolites helps to determine the plants' survival and development, but the creation of any blend is complicated by the fact that individual specialized metabolites can have contrasting effects on the plant. For example, individual specialized metabolites can defend the plant against some stressors while simultaneously making the plant more sensitive to other biotic or abiotic stresses (*Agrawal, 2000*; *Bialy et al., 1990*; *Erb and Kliebenstein, 2020*; *Futuyma and Agrawal, 2009*; *Hu et al., 2018*; *Lankau, 2007*; *Opitz*

**eLife digest** Since plants cannot move, they have evolved chemical defenses to help them respond to changes in their surroundings. For example, where animals run from predators, plants may produce toxins to put predators off. This approach is why plants are such a rich source of drugs, poisons, dyes and other useful substances. The chemicals plants produce are known as specialized metabolites, and they can change a lot between, and even within, plant species. The variety of specialized metabolites is a result of genetic changes and evolution over millions of years.

Evolution is a slow process, yet plants are able to rapidly develop new specialized metabolites to protect them from new threats. Even different populations of the same species produce many distinct metabolites that help them survive in their surroundings. However, the factors that lead plants to produce new metabolites are not well understood, and it is not known how this affects genetic variation.

To gain a better understanding of this process, Katz et al. studied 797 European variants of a common weed species called *Arabidopsis thaliana*, which is widely studied. The investigation found that many factors affect the range of specialized metabolites in each variant. These included local geography and environment, as well as genetics and population history (demography). Katz et al. revealed a pattern of relationships between the variants that could mirror their evolutionary history as the species spread and adapted to new locations.

These results highlight the complex network of factors that affect plant evolution. Rapid diversification is key to plant survival in new and changing environments and has resulted in a wide range of specialized metabolites. As such they are of interest both for studying plant evolution and for understanding their ecology. Expanding similar work to more populations and other species will broaden the scope of our ability to understand how plants adapt to their surroundings.

*and Müller, 2009*; *Uremıs et al., 2009*; *Züst and Agrawal, 2017*). These opposing effects create offsetting ecological benefits and costs for individual metabolites. Integrating these offsetting effects across dynamic environments involves multiple selective pressures that might contribute to shaping the genetic and metabolic variation within a species (*Fan et al., 2019*; *Kerwin et al., 2015*; *Malcolm, 1994*; *Sønderby et al., 2010*; *Szakiel et al., 2011*; *Wentzell and Kliebenstein, 2008*; *Züst et al., 2012*).

Significant advances have been made in recent decades to identify genetic sources contributing to metabolic variation. A common finding of these studies is that the metabolic variation within and between species is the result of structural variation at the enzymes responsible for the chemical structures, or variation at the expression levels of these enzymes, which contributes to the quantitative variation in specialized metabolism (*Chan et al., 2011*; *Chan et al., 2010*; *Fan et al., 2019*; *Kroymann et al., 2003*; *Moore et al., 2019*; *Schilmiller et al., 2012*). These structural and regulatory variants and the resulting chemical variation strongly influence plant fitness in response to a broad range of biotic interactions including herbivores and other plant species (*Bednarek and Osbourn, 2009*; *Brachi et al., 2015*; *Kerwin et al., 2017*; *Kerwin et al., 2015*; *Lankau and Kliebenstein, 2009*; *Lankau and Strauss, 2007*; *Lankau, 2007*). The potential for these genetic variants influencing plant chemical variation is derived from the enhanced proportion of gene duplication in enzyme encoding genes for specialized metabolism, both at the local and whole genome level (*Kliebenstein et al., 2001c*; *Moghe and Last, 2015*). Many mechanistic studies of natural variation in specialized metabolism have focused on biallelic phenotypic variation linked to loss-of-function variants. However, it is not clear if biallelic phenotypic variation is created by biallelic genetic causation when investigating a large collection of individuals from wide-ranging populations within a species.

If selective pressures are sufficiently non-linear, it is possible to have repeated and independent generation of structural variants creating the same metabolic variation in processes that are akin to parallel and convergent evolution are used to describe interspecific variation. Specifically, parallel and convergent evolution describe independent evolution of the same trait that differs depending on the beginning state of the organisms. In parallel evolution, the lineages begin from the same state and in parallel evolve to the same new state, while in convergent evolution the lineages start at different states and independently converge on the same new state (*Figure 1*). In this context, we are focusing

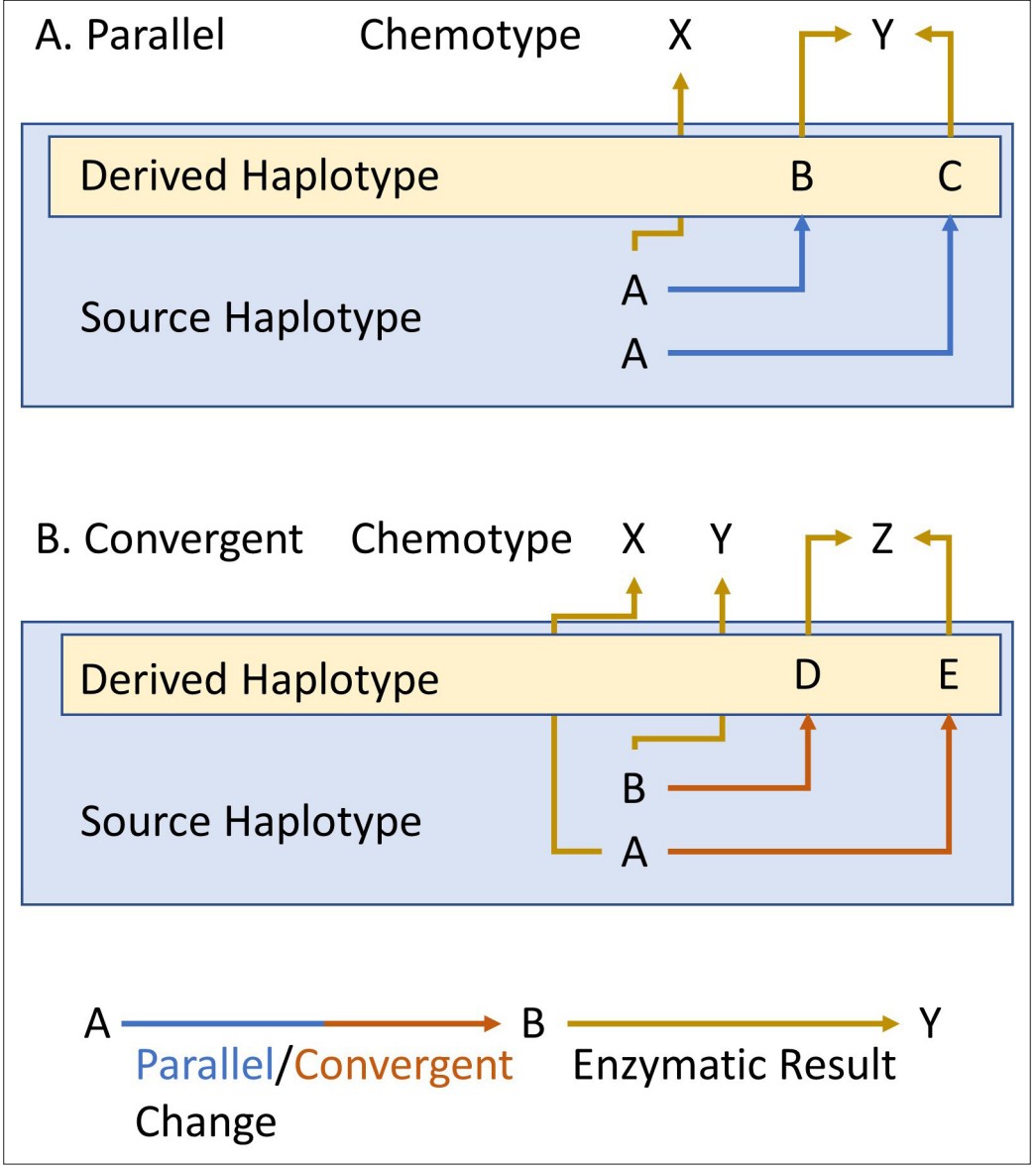

**Figure 1.** Parallel and convergent evolution. The schema describes our use of parallel (**A**) and convergent (**B**) evolution for within-species chemotypic variation. The letters in the blue box represent the state of the source/ancestral haplotypes. The letters within the yellow box represent the newly derived haplotypes that arose by genetic mutation in the source haplotype. Finally, X, Y and Z show the chemotypes that arise from each haplotype. Blue and red arrows represent parallel or convergent genetic changes (respectively), while mustard arrows represent the enzymatic result.

the analogy on the fact that the two processes differ where they begin, same or different state. This raises the possibility for chemical variation within a species to exhibit parallel evolution, wherein independent new haplotypes with identical metabolic consequences arise multiple times from single-core haplotype. Equally it may be possible to find within-species convergent evolution, where genotypes with the same metabolic profile actually contain completely different haplotypes that themselves arose from distinct haplotypic lineages. These genetic processes and interplay between genetics and selection overlap with neutral demographic processes like gene flow. Thus, it is necessary to understand how the intersection of environmental pressure, demography and genomic complexity gives rise to the pattern of metabolic variation across a plant species.

To better understand how genomic variation, demography and environmental pressure shape the variation of specialized metabolism within a species, we used the Arabidopsis glucosinolate (GSL)

pathway as a model. GSLs are a diverse class of specialized metabolites that display extensive variation across the order Brassicales, which includes the model plant Arabidopsis (*Arabidopsis thaliana*) (*Bakker et al., 2008*; *Benderoth et al., 2006*; *Brachi et al., 2015*; *Chan et al., 2010*; *Daxenbichler et al., 1991*; *Halkier and Gershenzon, 2006*; *Kerwin et al., 2015*; *Kliebenstein et al., 2001a*; *Kliebenstein et al., 2001b*; *Kliebenstein et al., 2001c*; *Rodman et al., 1981*; *Rodman, 1980*; *Sønderby et al., 2010*; *Wright et al., 2002*). GSLs consist of a common core structure with a diverse side chain that determines biological activity in defense, growth, development and abiotic stress resistance (*Beekwilder et al., 2008*; *Hansen et al., 2008*; *Hasegawa et al., 2000*; *Katz et al., 2020*; *Katz et al., 2015*; *Malinovsky et al., 2017*; *Salehin et al., 2019*; *Yamada et al., 2003*). The Arabidopsis-GSL system is an optimal model to study the species-wide processes driving specialized metabolite variation because the identity of the whole biosynthetic pathway is known, including the major causal loci for natural variation (*Benderoth et al., 2006*; *Brachi et al., 2015*; *Chan et al., 2011*; *Chan et al., 2010*; *Hansen et al., 2007*; *Kliebenstein et al., 2001a*; *Kliebenstein et al., 2002b*; *Kliebenstein et al., 2002a*; *Kroymann and Mitchell-Olds, 2005*; *Pfalz et al., 2007*; *Sønderby et al., 2010*; *Wentzell et al., 2007*). These major loci have been proven to influence Arabidopsis fitness and can be linked to herbivore pressure (*Brachi et al., 2015*; *Hansen et al., 2008*; *Jander et al., 2001*; *Kerwin et al., 2017*; *Kerwin et al., 2015*; *Züst et al., 2012*). Beyond the major causal loci, there is also evidence from genome-wide association (GWA) studies for highly polygenic variation in the genetic background that contributes to modulating GSL variation (*Chan et al., 2011*). The public availability of over 1000 widely distributed accessions with genomic sequences facilitates phenotyping GSL variation across a large spatial scale and analyses of causal haplotypes at the major GSL causal loci.

In Arabidopsis and other Brassicas, the main GSLs are methionine-derived, aliphatic, GSLs. Variation in the structure of aliphatic GSL is controlled by natural genetic variation at three loci: *GS-Elong*, *GS-AOP* and *GS-OH*. The specific alleles at these three loci combine to determine a predominant chemical structure and define chemically distinct aliphatic GSL chemotypes. In addition to these large-effect loci, there is a large suite of loci that can quantitatively alter the total accumulation and relative concentrations of GSLs within each chemotype (*Brachi et al., 2015*; *Chan et al., 2011*; *Chan et al., 2010*). *GS-Elong* differentially elongates the methionine side chain by the methylthioalkylmalate synthase enzymes (MAM). The elongation of the side chain by one methylene group is the result of one cycle that includes three steps: deamination of the methionine to create a $\omega$-methylthio-2-oxoalkanoic-acid, condensation of the $\omega$-methylthio-2-oxoalkanoic-acid with acetyl-CoA, and then isomerization and oxidative decarboxylation. The one carbon longer outcome can then undergo additional cycles of elongation (*Benderoth et al., 2006*; *Graser et al., 2000*; *Kroymann et al., 2001*; *Textor et al., 2007*). In Arabidopsis, MAM2 catalyzes the addition of one carbon to the side chain, creating GSLs with three carbon side chains. MAM1 catalyzes the addition of two carbons to make GSLs with four carbon side chains (*Figure 2*). MAM3 (also known as MAM-L) catalyzes the additions up to six carbons (*Kliebenstein et al., 2001c*; *Kroymann et al., 2003*; *Mithen et al., 1995*; *Textor et al., 2007*). The core pathway leads to the creation of methylthio GSL (MT). Then, the MT is converted to a methylsulfinyl (MSO) with a matching number of carbons (*Giamoustaris and Mithen, 1996*; *Hansen et al., 2007*). Structural variation at the *GS-AOP* locus leads to differential modification of the MSO by differential expression of a family of 2-oxoacid-dependent dioxygenases (2ODD). The AOP2 enzyme removes the MSO moiety leaving an alkenyl sidechain, while AOP3 leaves a hydroxyl moiety. Previous work has suggested three alleles of *GS-AOP*: the *OHP* allele that expresses only *AOP3* and accumulates terminal OH containing GLS in the leaves and seeds; an alkenyl allele expressing *AOP2* in the leaf and *AOP2* and *AOP3* in the seed leading to solely alkenyl GLS in the leaf and both alkenyl and OH aliphatic GLS in the seed; and a final allele containing a null mutation in the *AOP2* gene that accumulates MSO GLS in the leaf and enhanced MSO and OH GLS in the seed. (*Figure 2*; *Chan et al., 2010*; *Kliebenstein et al., 2001b*; *Kliebenstein et al., 2001c*; *Mithen et al., 1995*). The C4 alkenyl side chain can be further modified by adding a hydroxyl group at the 2C via the GS-OH 2-ODD (*Figure 2*; *Hansen et al., 2008*). In spite of the evolutionary distance, independent variation at the same three loci influences the structural diversity in aliphatic-GSLs within Brassica, Streptanthus and Arabidopsis (*Kliebenstein and Cacho, 2016*; *Lankau and Kliebenstein, 2009*). For example, the MAMs responsible for C3 GSLs in Arabidopsis and Brassica represent two independent lineages, same as the MAMs responsible for C4 GSLs; in fact, the *MAM* locus contains at least three independent lineages that recreate the same length variation (*Abrahams et al., 2020*). This indicates repeated evolution across

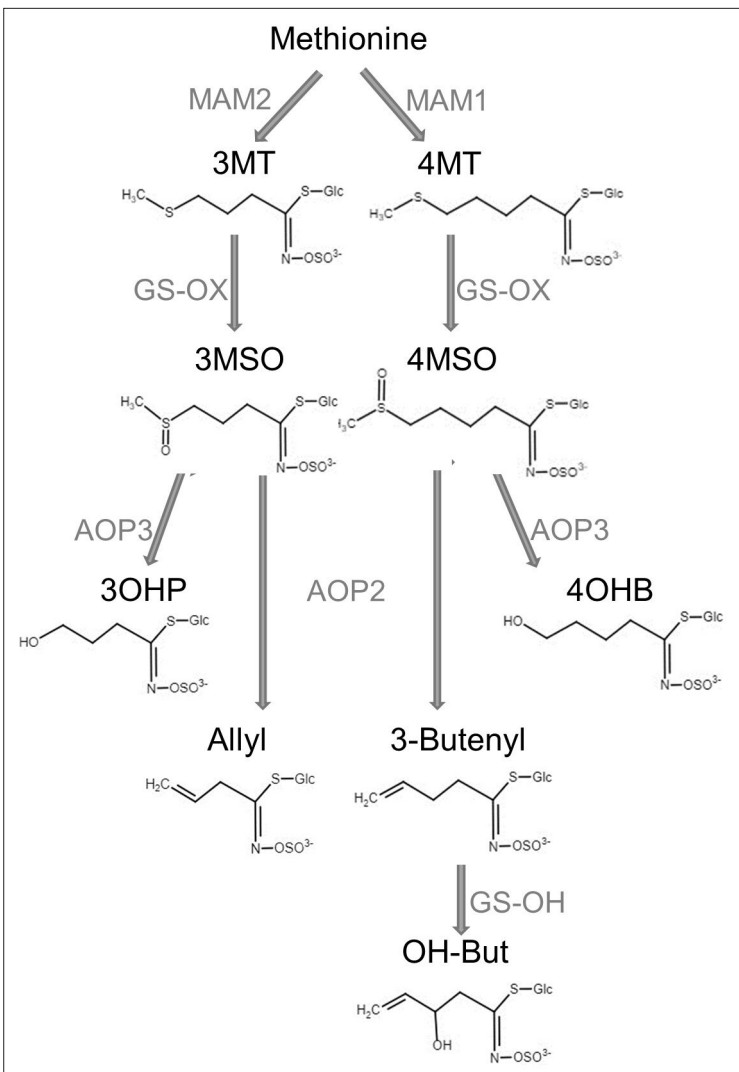

**Figure 2.** Aliphatic glucosinolate (GSL) biosynthesis pathway. Short names and structures of the GSLs are in black. Genes encoding the causal enzyme for each reaction (arrow) are in gray. *GS-OX* is a gene family of five or more genes. OH-But: 2-OH-3-Butenyl.

species, but it is not clear how frequently these loci are changing within a single species or how ecological or demographic processes may shape within-species variation at these loci.

In this work, we described GSL variation in seeds of a collection of 797 *A. thaliana* natural accessions collected from different locations mainly in and around Europe. The amounts of GSLs can vary across different tissues and life stages, but there is a strong correlation in the type of aliphatic GSL produced across tissues (*Brown et al., 2003*; *Kliebenstein et al., 2001a*; *Kliebenstein et al., 2001b*; *Petersen et al., 2002*). Thus, in most cases the chemotype of the seeds is the same as the leaves. The seeds have the highest level of GSLs in Arabidopsis and are stable at room temperature until germination, which makes the seeds a perfect tissue to survey variation. Further, GSLs are known to be important for seed defenses against herbivores and pathogens (*Raybould and Moyes, 2001*). By combining GSL seed measurements with prior whole-genome sequencing in a European collection of accessions, we show that all three major causal loci controlling GSL metabolic diversity contain multiple independently derived alleles that recreate the same phenotypes using a combination of single nucleotide polimorphism (SNPs) and structural variation. Using these causal genotypes and chemotypes in combination with their geographic distribution provided evidence that the distribution of GSL metabolic diversity across Europe is influenced by a combination of demography and ecological factors. The ecological relationships to chemotype suggested a potential for variation in selective

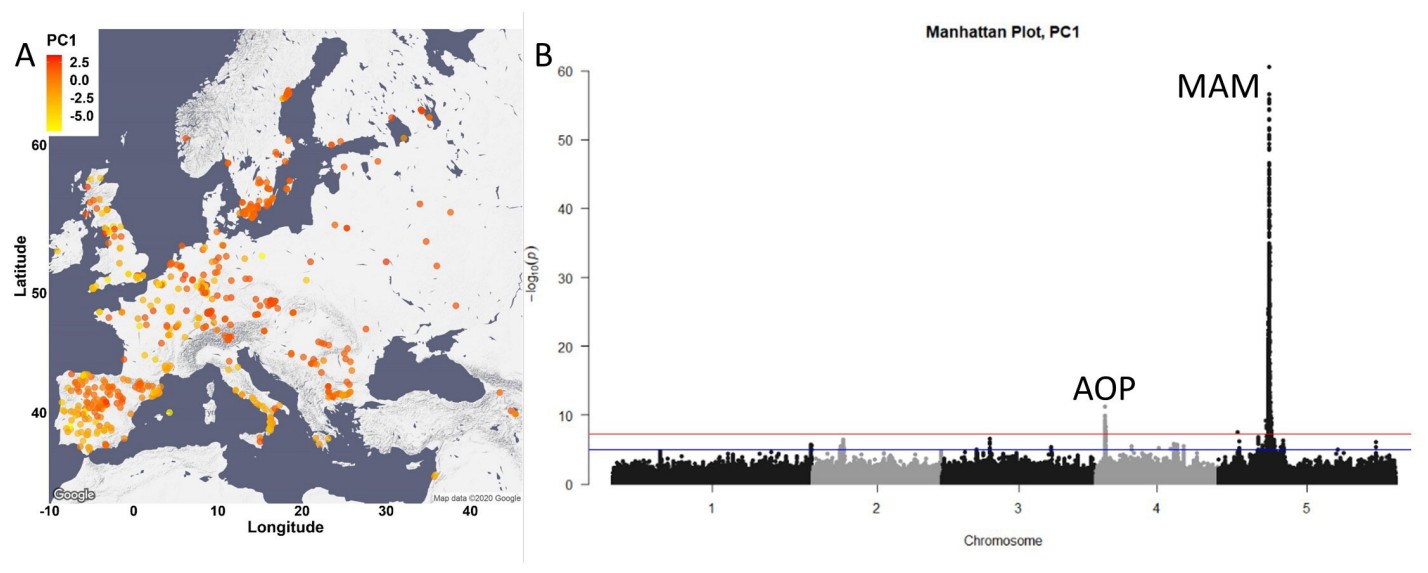

**Figure 3.** Glucosinolate variation across Europe is dominated by two loci. (**A**) The accessions are plotted on the map based on their collection site and colored based on their principal component (PC)1 score. (**B**) Manhattan plot of genome-wideassociation analyses using PC1. Horizontal lines represent 5% significance thresholds using Bonferroni (red) and Benjamini–Hochberg (blue).

The online version of this article includes the following figure supplement(s) for figure 3:

**Figure supplement 1.** Glucosinolate (GSL)-based principal component (PC) analysis.

**Figure supplement 2.** Glucosinolate variation across Europe is dominated by two loci.

**Figure supplement 3.** Manhattan plots of genome-wideassociation performed based on individual glucosinolate amounts as traits.

processes across the geographic regions studied. Future work will be needed to identify the specific biotic and/or abiotic factors shaping this distribution.

## Results

### GSL variation across Europe

To investigate the genetic, environmental and demographic parameters influencing the distribution of Arabidopsis GSL chemotypes, we measured GSLs from seeds of a collection of 797 *A. thaliana* natural accessions ( *The 1001 Genomes Consortium, 2016*). These Arabidopsis accessions were collected from different geographical locations, mainly in and around Europe (*Figure 3A*). 23 different GSLs were detected and quantified, identifying a wide diversity in composition and amount among the natural accessions with a median heritability of 83%, ranging from 34% to 93% (*Supplementary file 1*). To summarize the GSL variation among the accessions, we performed principal component analyses (PCA) on the accumulation of all the individual GSLs across the accessions as an unbiased first step. The first two PCs only captured 33% of the total variation with PC1 describing GSLs with four and seven carbons and PC2 mainly capturing GSLs with eight carbons in their side chain (*Figure 3— figure supplement 1*). Previous work using a collection of predominantly central European accessions had suggested a simple continental gradient chain-elongation variation from the south-west (SW) (that is enriched with alkenyl and hydroxyalkenyl GSLs) to the north-east (NE) (*Brachi et al., 2015*; *Züst et al., 2012*). To assess if this was still apparent in this larger collection, we plotted the accessions based on their geographical locations and colored them based on their PC1 and PC2 scores that are linked to chain elongation variation (*Figure 3A* and *Figure 3—figure supplement 2A*, respectively). This larger collection shows that there is not a single gradient shaping GSL diversity across Europe (*Figure 3A*). Instead, the extended sampling of accessions around the Mediterranean Basin in this collection shows that the SW to NE pattern reiterates within the Iberian Peninsula. In each of these areas (Iberian Peninsula and Central Europe), the SW is enriched with C4 GSLs, and the NE with C3 GSLs (*Figure 3—figure supplement 1*).

To test which of the major causal loci are detectable in this collection and to identify new genomic regions that are associated with the observed GSL variation, we performed GWA (with EMMAX algorithms) analyses using the PC1 and PC2 values. This collection of natural accessions presents a dense variant map that is 3× larger than previous GSL GWA mapping populations and includes 6,973,565 SNPs. In spite of the large population size, both PC1- and PC2-based analyses identified the same two major peaks covering two of the known causal gene clusters controlling GSL diversity (*Figure 3B* for PC1 GWA analyses, *Figure 3—figure supplement 2B* for PC2 GWA analyses) (*Brachi et al., 2015*; *Chan et al., 2011*; *Chan et al., 2010*). The largest peak in both cases is the *GS-Elong* locus on chromosome 5, containing the *MAM1* (AT5G23010), *MAM2* (that is not present in Col-0 plants) and *MAM3* (AT5G23020) genes.

The peak on chromosome 4 is the *GS-AOP* locus containing the *AOP2* and *AOP3* genes (AT4G03060 and AT4G03050, respectively). Applying a more permissive cutoff did not result in the detection of any other related genes (*Supplementary file 2*). Previous QTL mapping and molecular experiments have shown that the genes within *GS-AOP* and *GS-Elong* loci are the causal genes for GSL variation within these regions (*Benderoth et al., 2006*; *Brachi et al., 2015*; *Chan et al., 2011*; *Chan et al., 2010*; *Kliebenstein et al., 2001a*; *Kliebenstein et al., 2002a*; *Kliebenstein et al., 2002a*; *Kroymann and Mitchell-Olds, 2005*; *Pfalz et al., 2007*; *Wentzell et al., 2007*). Surprisingly, none of the other known natural variants within the GSL biosynthetic pathway (listed in *Supplementary file 2*) were identified by GWA including three that were found with 96 accessions and three that were found with 595 accessions using PC1 and 2 (*Brachi et al., 2015*; *Chan et al., 2011*; *Chan et al., 2010*; *Kliebenstein, 2009*). Performing GWA studies using the accumulation of each of the 23 individual GSL detected in this collection resulted in an identical result, no additional known GSL-related genes were detected, while a few additional unknown genes were found (*Figure 3—figure supplement 3* and *Supplementary file 2*). One explanation for that is that the dense sampling in this collection is available for mainly the Iberian Peninsula, the southern coast of Sweden and the south-western coast of Italy, and is still insufficient for Central Europe. Another possibility is that allelic heterogeneity for the other loci, and more complex patterns of interaction, may hamper their detection and influenced this high false-negative error rate where ~80% of prior validated natural variants found using multiple RIL populations were missed.

## Complex GSL chemotypic variation

One potential complicating factor is that GSL chemotypic variation is best described as a discrete multimodal distribution involving the epistatic interaction of multiple genes which PCA's linear decomposition cannot accurately capture (*Figure 2*). To test if PCA was inaccurately describing GSL chemotypic variation, we directly called the specific GSL chemotypes in each accession. Using Arabidopsis QTL mapping populations and GWA, we have shown that the *GS-AOP*, *Elong* and *OH* loci determine seven discrete chemotypes, 3MSO, 4MSO, 3OHP, 4OHB, Allyl, 3-Butenyl, 2-OH-3-Butenyl (*Figure 2*), that can be readily assigned from GSLs' phenotypic data (*Brachi et al., 2015*; *Chan et al., 2011*; *Chan et al., 2010*; *Kliebenstein et al., 2001a*). The presence and amounts of these seven chemotypes provide a reliable indication about the existence and activity of each of the major GSL loci. Using accessions with previously known chemotypes and genotypes, we developed a phenotypic classification scheme to assign the chemotype for each accession (*Figure 4*; for details, see Methods and *Figure 4—figure supplements 1–3*; for structures, see *Figure 2* and *Supplementary file 1*). Since the aliphatic GSLs' composition in the seeds reliably indicates the GSL structural composition in the other plant's life stages and tissues, assigning a chemotype for each accession based on the seeds' composition is expected to be highly stable across tissues of the same accession (*Brown et al., 2003*; *Chan et al., 2011*; *Chan et al., 2010*; *Kliebenstein et al., 2001a*; *Kliebenstein et al., 2001b*). Most accessions were classified as 2-OH-3-Butenyl (27%) or Allyl (47%) with lower frequencies for the other chemotypes. Mapping the chemotypes onto Europe showed that the PCA decomposition was missing substantial information on GSL chemotype variation (*Figure 4*). Instead of a continuous distribution across Europe, the chemotype classifications revealed specific geographic patterns. Central and parts of Northern Europe (like north Germany and Poland) were characterized by a high variability involving the co-occurrence of individuals from all chemotypes. In contrast, southern Europe, which presents a dense sampling, including the Iberian Peninsula, Italy and the Balkan, has two predominant chemotypes, Allyl or 2-OH-3-Butenyl, that are separated from each other by a clear and sharp

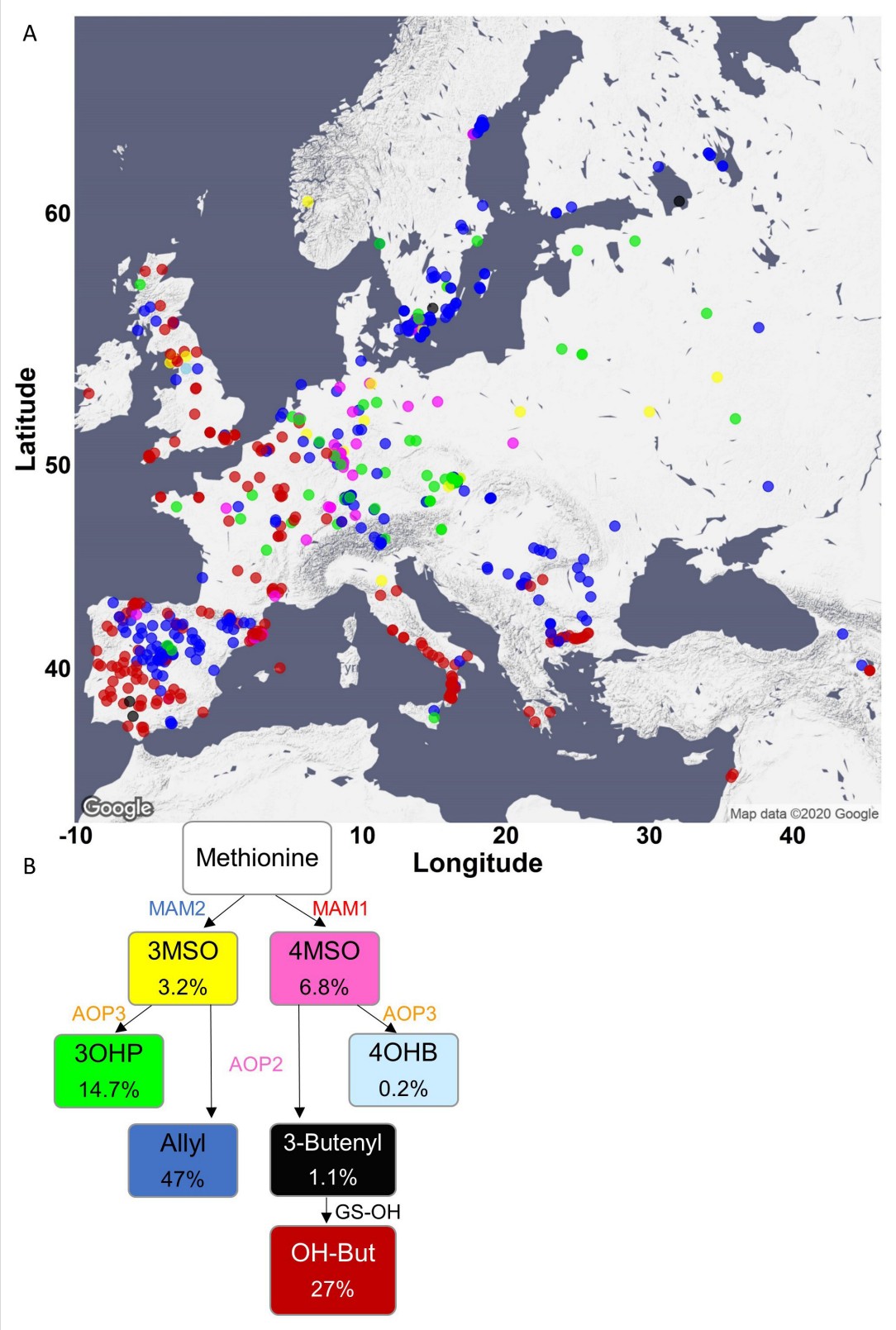

**Figure 4.** Phenotypic classification based on glucosinolate (GSL) content. (**A**) Using the GSL accumulation, each accession was classified to one of seven aliphatic short-chained GSL chemotypes based on the enzyme functions as follows: *MAM2, AOP* null: classified as 3MSO dominant, colored in yellow. *MAM1, AOP* null: classified as 4MSO dominant, colored in pink. MAM2, AOP3: classified as 3OHP dominant, colored in green. *MAM1, AOP3*: classified as 4OHB dominant, colored in light blue. *MAM2, AOP2*: classified as Allyl dominant, colored in blue. *MAM1, AOP2, GS-OH* non-functional: classified

*Figure 4 continued on next page*

*Figure 4 continued*

as 3-Butenyl dominant, colored in black. *MAM1, AOP2, GS-OH* functional: classified as 2-OH-3-Butenyl dominant, colored in red. The accessions were plotted on a map based on their collection sites and colored based on their dominant chemotype. (**B**) The coloring scheme with functional GSL enzymes in the aliphatic GSL pathway is shown with the percentage of accessions in each chemotypes (out of the total 797 accessions) shown in each box.

The online version of this article includes the following source data and figure supplement(s) for figure 4:

**Source data 1.** Environmental conditions differentially associate with *MAM* status in the north versus the south.

**Figure supplement 1.** Phenotypic classification based on the dominant *MAM* enzyme.

**Figure supplement 2.** Phenotypic classification based on the dominant *AOP* enzymes.

**Figure supplement 3.** Phenotypic classification based on *GS-OH* enzyme activity.

**Figure supplement 4.** Geographic partitioning of the collection.

geographic partitioning (*Figure 4* and *Figure 4—figure supplement 4*). Uniquely, Swedish accessions displayed a striking presence of almost solely Allyl chemotypes. Deeper sampling is required to test if this is or is not mirrored on the eastern side of the Baltic Sea as the few accessions from that region are almost solely 3OHP chemotypes (Finnish, Lithuanian, Latvian or Estonian accessions). Directly assigning GSL variation by discrete chemotypes provided a more detailed image not revealed by PCA decomposition. Further, the different chemotypic to geographic patterns suggest that there may be different pressures shaping GSL variation particularly when comparing Central and Southern Europe.

## Geography and environmental parameters affect GSL variation

Because GSL chemotypes may be more reflective of local environment, we proceed to test if they are associated with weather parameters and landscape conditions. Further, given the difference in chemotype occurrence in Central and Southern Europe, we hypothesized that these environmental connections may change between Central and Southern Europe. For these tests, we chose environmental parameters that capture a majority of the environmental variance and by that may describe the type of ecosystem (*Ferrero-Serrano and Assmann, 2019*). We assigned each accession the environmental value based on its location. These environmental parameters include geographic proximity (distance to the coast), precipitation descriptors (precipitation of wettest and driest month) and temperature descriptors (maximal temperature of warmest month and minimal temperature of coldest month) and capture major abiotic pressures as well as provide information about the type of ecosystem in which each accession exists. Because demography and environment can be confounded, we included demography in our models using the previously assigned genomic groupings as components of the model (*The 1001 Genomes Consortium, 2016*). Further, we included specific geographic information by assigning the accessions to a northern or a southern collection, based on their location in relation

**Table 1.** Environmental conditions differentially associate with major chemotypes across geographic location.

Linear model for the two major chemotypes, Allyl and 2-OH-3-Butenyl, was conducted with the indicated environmental parameters, for the northern and southern collection, separately (for more details, see Methods). The table shows p values for each term from the linear model. For the interaction with geography, the linear model was run using the total dataset, and the geography parameter (north or south) was added to the model.

| Environmental parameter | Effect on chemotype – north | Effect on chemotype – south | Interaction with geography |
|---|---|---|---|
| Genomic group | <0.0001 | <0.0001 | <0.0001 |
| Max temperature of warmest month | 0.0382 | <0.0001 | 0.3574 |
| Min temperature of coldest month | 0.0007 | <0.0001 | 0.0049 |
| Precipitation of wettest month | 0.1645 | 0.0003 | 0.0094 |
| Precipitation of driest month | 0.0665 | 0.2026 | 0.47425 |
| Distance to the coast | 0.2781 | 0.02680 | 0.1279 |

to the following chain of mountains: the Pyrenees, the Alps and the Carpathians (*Figure 4—figure supplement 4*). We then ran a linear model for each geographic area separately (north and central vs. south) to check if the environmental parameters and the genomic population group associate with specific chemotypes. To directly test for an interaction of environment and geography, we ran the model with all accessions and incorporated the geography parameters and genomic population group. As the most frequent chemotypes in the collection are Allyl and 2-OH-3-Butenyl (*Figure 4—figure supplement 4B*), we focused the models on these chemotypes. The models showed that the environmental conditions have different relationships to the chemotypes that shift by geographic areas. Moreover, two of the parameters (min temp of coldest month and precipitation of wettest month) have a significant interaction with geography, suggesting that the relationship of these environmental parameters to specific GSL chemotypes is different between Northern and Southern Europe (*Table 1*; for details on the models, see Methods). This suggests that the relationship of GSL chemotype to environmental parameters varies across geographic regions of Europe rather than fitting a simple linear model.

As the two main chemotypes in the collection differ by the length of the carbon chain (C3 for Allyl, C4 for 2-OH-3-Butenyl), we created a linear model to further check the interaction between each environmental condition to geography in respect to the carbon chain length. As was shown by the chemotypes models, most of the environmental parameters (min temp of coldest month, precipitation of wettest month and distance to the coast) significantly interacted with geography, showing again that the relationship of environment to GSL alleles changes across Europe (*Figure 4—source data 1*; for details on the models, see Methods). Conducting this analysis for each of the geographic areas separately highlighted this by showing that these parameters have different effects on the carbon chain length in each of the areas (*Figure 4—source data 1*).

## The genetic architecture of GSL variation

The presence of different GSL chemotype to environmental relationships across Europe raises the question of how these chemotypes are generated. Are these chemotypes from locally derived alleles or obtained by the intermixing of widely distributed causal alleles? Further, if there are multiple alleles, do they display within-species convergent or parallel signatures? We focus on the *GS-AOP*, *GS-Elong* and *GS-OH* loci, the causal genes creating Arabidopsis GSL chemotypes, and use the available genomic sequences in all these accessions to investigate the allelic variation in these genes to map the allelic distribution and test the potential for convergent and/or parallel evolution within each locus.

*GS-Elong*: Because the variation in the *GS-Elong* locus is caused by complex structural variation in *MAM1* and *MAM2* that is not resolvable using the available data from short-read genomic sequence, we used the *MAM3* sequence within this locus to ascertain the genomic relationship of accessions at the causal *GS-Elong* locus (*Kroymann et al., 2003*). We aligned the *MAM3* sequence from each of the accessions, rooted the tree with the *Arabidopsis lyrata* orthologue (*MAMb*) and colored the tree tips based on the accessions-dominant chemotype.

The accessions were distributed across eight distinctive clades with each clade clustering accessions having either a C3 or C4 phenotype (*Figure 5A* and *Figure 5—figure supplement 1* for bootstrap support). The clades C3/C4 status altered across the tree with three of the clades expressing C3 (*MAM2*) and five clades expressing the C4 (*MAM1*). The use of *MAM3* clades as proxy for C3/C4 GLS chemotypes is supported by prior genomic sequencing of the *GS-Elong* region from 15 accessions (*Figure 5B*; *Kroymann et al., 2003*). To test for potential within locus recombination that may influence the overarching patterns, we compared the *MAM3* tree to a tree obtained using *MYB37*, which is on the opposite end of the *MAM* locus from *MAM3* (*Figure 5—figure supplement 1F*). We found that while the order of the clades in the *MYB37* tree is different than their order in the *MAM3* tree, the accessions' classification to clades was similar among the two trees. This suggests that while there are potentially individual instances of within-locus recombination they are not influencing the overall genotype to chemotype linkage from *MAM3*, and *MAM3* can be used as a reliable reflection of the structural variation in this locus.

Six of the clades in the *MAM3* tree include accession/s with a previously sequenced *MAM* locus (*Figure 5B*; *Kroymann et al., 2003*), while two clades (clades 6 and 7) did not include any accession with a previously determined structure. We obtained long-read-based sequencing of 11 additional accessions from the 1001 Genome project for the *MAM* locus that included accessions in all clades

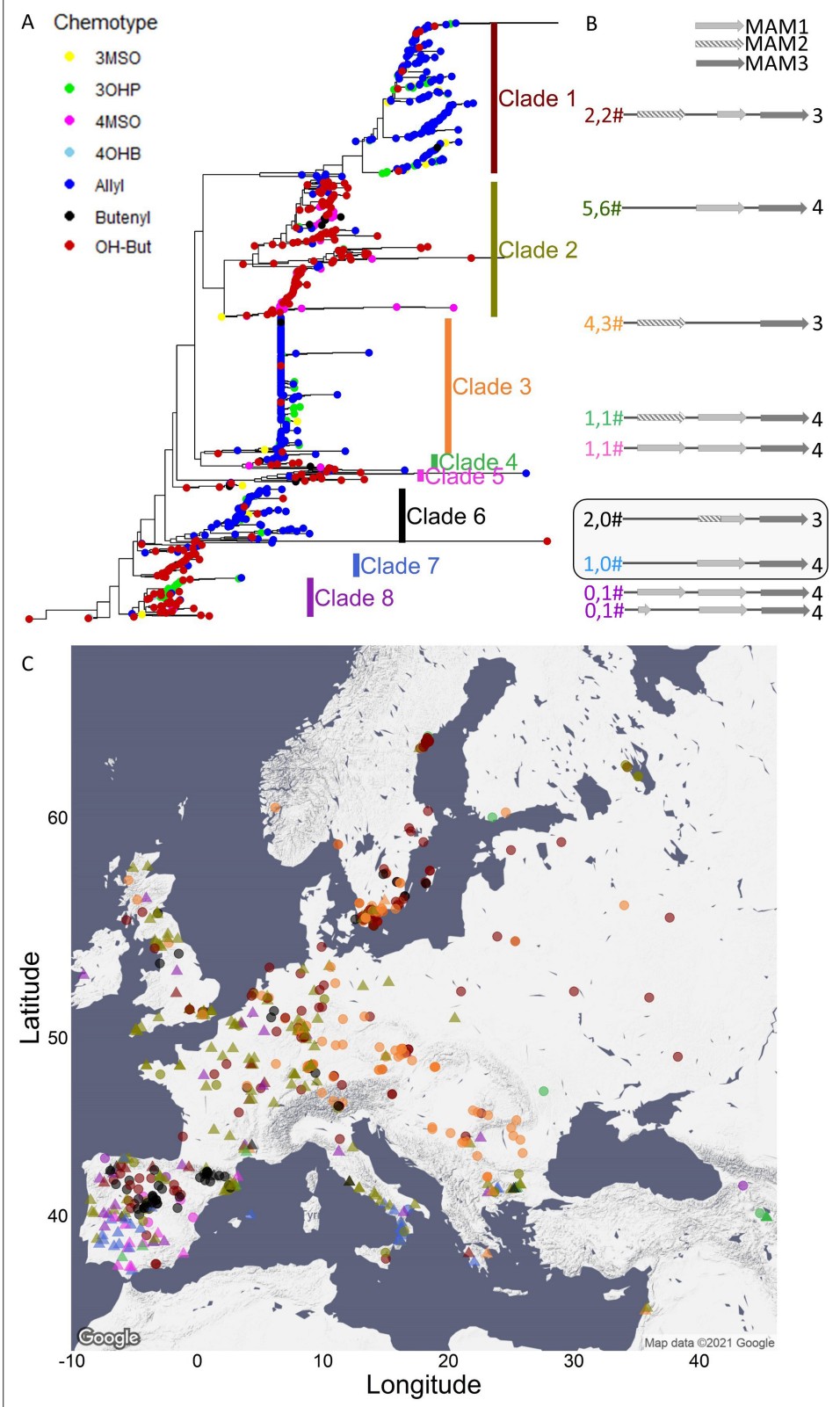

**Figure 5.** *MAM3* phylogeny. (**A**) *MAM3* phylogeny of *Arabidopsis thaliana* accessions, rooted by *Arabidopsis lyrata* *MAMb*, which is not shown because of distance. Tree tips are colored based on the accession chemotype. (**B**) The genomic structure of the *GS-Elong* regions in the previously sequenced accessions is shown based on ***Kroymann et al., 2003***. The structures in the box are based on sequences obtained in this work. The numbers left to the

*Figure 5 continued on next page*

*Figure 5 continued*

structures indicate the number of sequenced accessions in this work (left) or by *Kroymann et al., 2003* (right). The numbers are colored based on their clades. Bright gray arrows represent *MAM1* sequences, and dashed arrows represent *MAM2* sequences. Dark gray arrows represent *MAM3* sequences. The number to the right of the genomic cartoon represents the number of carbons in the side chain. (**C**) Collection sites of the accessions colored by their clade classification (from section A) and shaped based on the side chain length of the aliphatic short-chained glucosinolates (circles for C3, triangles for C4).

The online version of this article includes the following source data and figure supplement(s) for figure 5:

**Figure supplement 1.** Support for the *MAM3* tree clades classification.

**Figure supplement 2.** Genomic structure of the GS-Elong regions.

**Figure supplement 2—source data 1.** Sequences of *MAM* locus.

**Figure supplement 3.** *MAM2* is an Arabidopsis thaliana specific gene.

**Figure supplement 4.** Iberia Peninsula presents low phenotypic variability and high genetic variation.

**Figure supplement 5.** Geographic distribution of *MAM* haplotypes.

including clades 6 and 7 (*Figure 5—figure supplement 2—source data 1* for sequences). This showed that clade 6 are accessions that have a haplotype that contains a previously described chimeric *MAM* gene that combines the 5′ of *MAM2* with the 3′ of *MAM1* (*Figure 5B* and *Figure 5—figure supplement 2*; *Benderoth et al., 2006*; *Kroymann et al., 2003*). In these accessions, the chimeric gene leads to predominantly C3 GSLs. Clade 7 has a haplotype that is highly similar to clade 2 with a single copy of *MAM1* leading to C4 GSLs. Comparing transposable elements in the two clades shows that they are different configurations.

The new sequenced accessions present in the clades with existing genomic haplotypes predominantly agreed with these previously published haplotypes. There were only three accessions with differences, two with a local duplication of a truncated *MAM1* pseudogene in clades 1 and 2 (PHW-34 and TAL 07, respectively), and a second with a local duplication of a *MAM1* pseudogene in clade 2 (Qar-8a, *Figure 5—figure supplement 2*).

The bootstrap support and smaller trees raised the possibility that clade 2 could be considered as two distinct clades (*Figure 5—figure supplement 1*, clades 2a and 2b). The chemotypes and haplotype in the accessions do not provide a clear mechanistic basis for separating this clade into two (*Kroymann et al., 2003*; *Figure 5*). Comparing the accessions across the main split in this clade suggested that one group of accessions (clade 2b) has lower total GSLs and a higher fraction of short-chain GSLs in comparison to the longer chain structures. Future work involving populations solely focused on this question would be needed to resolve the mechanistic basis of this difference and if this represents two distinct *MAM* loci.

One complication in interpreting the potential for parallel vs. convergent evolution in this locus is that the relationship between the major chemotype/haplotype groups is not resolvable with very low bootstraps (*Figure 5—figure supplement 1*). Functional parsimony would suggest that clade 4, by having both *MAM1* and *MAM2*, may represent a single haplotype that can give rise to the other functional haplotypes via independent mutations akin to parallel evolution. Supporting this potential is the observation that *MAM1* and *MAM2* are likely derived via a tandem duplication with ensuing divergence since the separation from *A. lyrata* (*Figure 5—figure supplement 3*; *Benderoth et al., 2009*; *Benderoth et al., 2006*). Fully resolving this would require collecting more accessions to identify additional alleles that may contain the information necessary to better resolve the relationships amongst the haplotypes.

Using this phylogeny, we investigated the presence of the different *GS-Elong* haplotypes across Europe to ask if each region has a specific allele/clade or if the alleles are distributed across the continent. Specifically, we were interested if the strong C3/C4 partitioning in Southern Europe was driven by the creation of local alleles or if this partitioning might contain a wide range of alleles. If the latter is true, this can argue for a selective pressure shaping this C3/C4 divide. To understand the patterning of the C3/C4 haplotypes and chemotypes in Iberia, we plotted the accessions on the map and colored them based on their *GS-Elong* clade (*Figure 5C* and *Figure 5—figure supplement 4*). As expected given that genetic variation in Iberia results from a series of range expansions from Central Europe and Africa (*Lee et al., 2017*; *Durvasula et al., 2017*), there is extensive mixing of

nearly all major European *GS-Elong* haplotypes in Iberia, except of clade 3 that is not present. In contrast, there is a sharp partitioning between the C3/C4 chemotypes created by these haplotypes. The strong geographic separation between the two chemotypes involving nearly all causal haplotypes (*Figure 5—figure supplement 4*) raises the possibility that the strong geographic partitioning of the C3/C4 chemotypes in Iberia may be driven by selective pressure enhancing the partitioning of the chemotypes, and not solely neutral demographic processes. The presence of a few accessions in Iberia that disagree with the sharp C3/C4 partition (*Figure 5—figure supplement 4A*) suggests that a new configuration of this loci arose in this area and is reflected in a few accessions. However, this requires further assessment.

Shifting focus to all of Europe showed that while most clades were widely distributed across Europe there were a couple of over-arching patterns (*Figure 5C* and *Figure 5—figure supplement 5*). *GS-Elong* clades 1 and 6 provide an example of potential gene flow between Iberia and Central Europe. In contrast, the absence of clade 3 in Iberia is more parsimonious with this haplotype having a glacial refugium in the Balkans followed by a northward flow wherein it mixed with the other clades. Other clades do not present evidence of a gene flow to the north as they are exclusive to the south as shown by clades 5 and 7. While these are both C4 clades, other C4 clades like clades 2 and 8 present a case of a gene flow to the north (*Figure 5—figure supplement 5*). This suggests that there are either differences in their GSL chemotype influencing their distribution or there are neighboring genes known to be under selection in Arabidopsis like *FLC* (AT5G10140) that may have influenced their distribution. In combination, this suggests that a complex demography is involved in shaping the chemotype's identity with some regions, Iberia, showing evidence of local selection while other regions, Central Europe, possibly showing a blend requiring further work to delineate (*Figure 5—figure supplement 5*).

*GS-AOP*: Side chain modification of the core MSO GSL is determined by the *GS-AOP* locus. Most of the accessions contain a copy of *AOP2* and a copy of *AOP3*, but only one of them will be functionally expressed (*Chan et al., 2010*), while in some cases both will be non-functional. To better understand the demography and evolution of the *GS-AOP* locus, we separately aligned the *AOP2* and *AOP3* sequences, rooted each tree with the *A. lyrata* orthologue and colored the trees tips based on the accessions-dominant chemotype (*Figure 6—figure supplement 1*).

The phylogenetic trees shared a very similar topology, yielding a clear separation between alkenyl (*AOP2* expressed) and hydroxyalkyl (*AOP3* expressed) accessions. Alkenyl expressing accessions like Cvi-0 with an expressed copy of the AOP2 enzyme formed a single continuous cluster (*Figure 6A* and *Figure 6—figure supplement 1*). In contrast, hydroxyalkyl (*AOP3* expressed) accessions clustered into two separate groups with one group of 3OHP-dominant accessions partitioning from the rest of the accessions at the most basal split in the tree (*Figure 6—figure supplement 1A*, *AOP2* tree). This haplotype is marked by having an inversion swapping the *AOP2* and *AOP3* promoters as shown in bacterial artificial chromosome sequencing of the Ler-0 accession (*Figure 6D*; *Chan et al., 2010*). The *AOP3* tree also identified a second group of 3OHP-dominant accessions located among the alkenyl accessions. Analyzing the sequences of these accessions reveals that this small group of 3OHP accessions has a complete deletion of *AOP2* and contains only *AOP3* (*Figure 6E*). Thus, there are at least two independent transitions from alkenyl to hydroxyalkyl GSLs within Arabidopsis, neither of which are related to the alkenyl to hydroxyalkyl conversion within *A. lyrata*. This indicates that there are multiple alkenyl to hydroxyalkyl GSL conversions both within and between Arabidopsis species.

The null accessions (MSO-dominant chemotypes) were identifiable in all the major clades on the tree (*Figure 6—figure supplement 1*, middle column of heatmap), suggesting that there are independent LOF mutations that abolish either *AOP2* or *AOP3*. Deeper examination of the sequences of these accessions identified three convergent LOF alleles leading to the MSO chemotype. Most of the null accessions harbor a 5 bps deletion in their *AOP2* sequence, which causes a frameshift mutation. This mutation arose within the alkenyl haplotype and was first reported in the Col-0 reference genome (*Figure 6B*; *Kliebenstein et al., 2001c*). In addition, there are additional independent LOF events arising in both the alkenyl haplotype (e.g., Sp-0, *Figure 6C*) and within the Ler-0 inversion haplotype (e.g., Fr-2, *Figure 6F*). Thus, *GS-AOP* has repeated LOF alleles arising within several of the major *AOP* haplotypes, suggesting convergent evolution of the MSO chemotype out of both the alkenyl and hydroxyalkyl chemotypes.

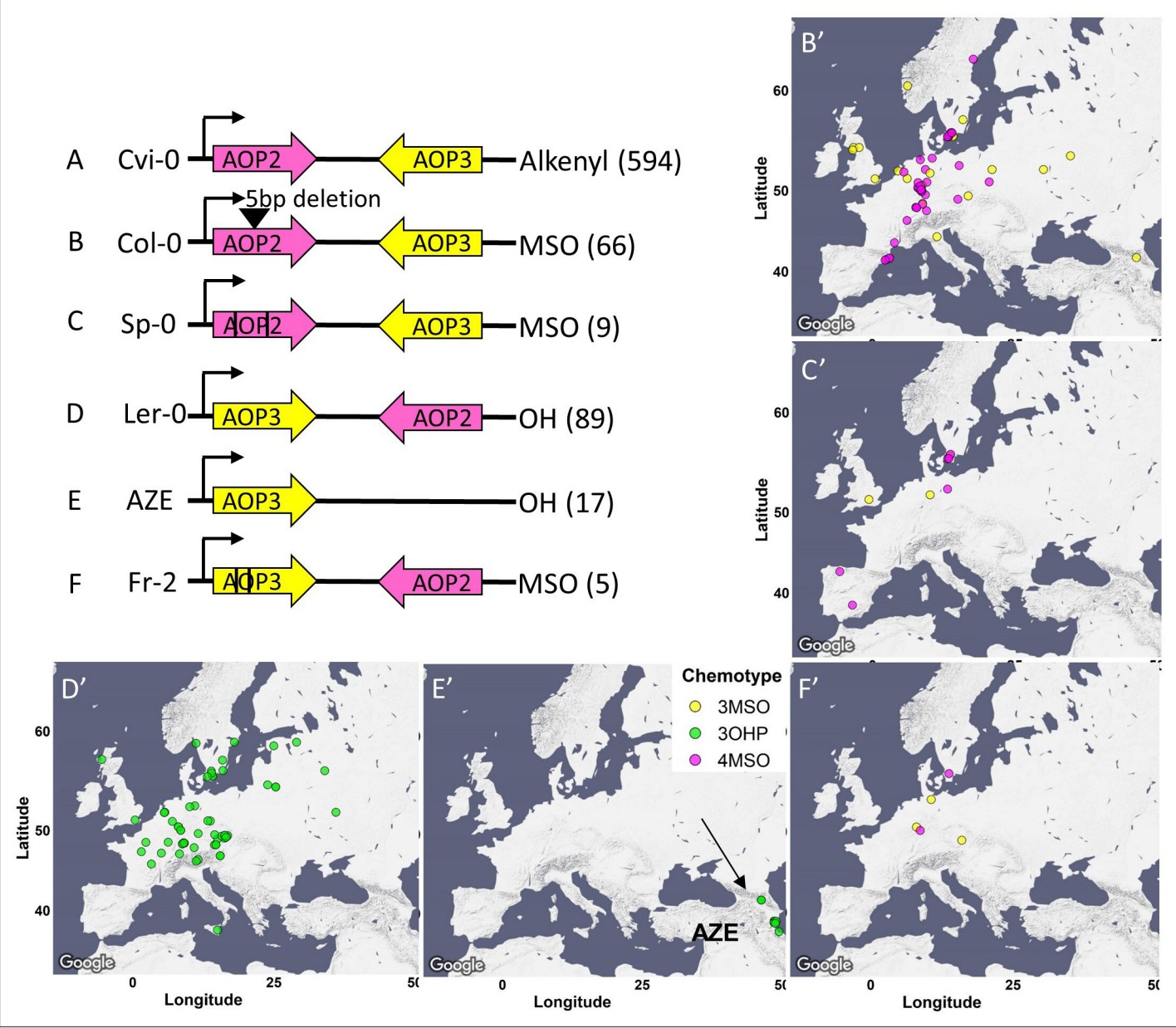

**Figure 6.** *AOP* genomic structure. The genomic structure and causality of the major *AOP2/AOP3* haplotypes are illustrated. Pink arrows show the *AOP2* gene while yellow arrows represent *AOP3*. The black arrows represent the direction of transcription from the *AOP2* promoter as defined in the Col-0 reference genome. Its position does not change in any of the regions. **A-F** represent the different structures. The black lines in **C** and **F** represent theoretical positions of independent variants creating premature stop codons. The GSL chemotype for each haplotype is listed to the right with the number of the accessions in brackets. The maps show the geographic distribution of the accessions from each structure.

The online version of this article includes the following figure supplement(s) for figure 6:

**Figure supplement 1.** *AOP* phylogeny.

Using the combined chemotype/genotype assignments at *GS-AOP*, we investigated the distribution of the alleles across Europe. The alkenyl haplotype is spread across the entire continent. In contrast, the hydroxyalkyl haplotypes are geographically more restricted. The Ler-like 3OHP haplotype is present in only Central and North Europe (*Figure 6D*), while the other 3OHP haplotype, possessing only *AOP3*, is limited to Azerbaijan, along the Caspian Sea (*Figure 6E*). In contrast to the distinct hydroxyalkyl locations, the distribution of the independent LOF null haplotypes overlaps with all of them being located within Central and North Europe (*Figure 6B, C and F*). The fact that these

**Table 2.** *GSOH* structure.

The structures of *GS-OH* in the 3-Butenyl accessions are illustrated. Gray boxes represent exons, and blue lines represent introns. Black line represents a mutation, and gray lines represent unknown lesions in hypothetical locations. The above mutations create non-functional *GS-OH* alleles. The fractions of the mutated accessions were calculated out of the total number of 3-Butenyl and OH-3-Butenyl accessions. The observed frequencies were calculated as the ratio between number of accessions with the specific mutation in non-C4 Alkenyl accessions and all the non-C4 Alkenyl accessions, as mentioned in the parentheses.

| Accession | Type of mutation | Allele structure | Fraction (out of C4 Alkenyl accessions) | Observed frequency (out of non-C4 Alkenyl accessions) |
|---|---|---|---|---|
| Sorbo, Pien | Polymorphism at SNP10831302 | | 0.009 (2/226) | 0.067 (38/564) |
| Cvi-0 | Active site mutation | | 0.004 (1/226) | 0.025 (14/564) |
| IP-Mot-0, IP-Tri-0 | Gene deletion | | 0.009 (2/226) | 0.055 (31/564) |
| Multiple accessions (T670, FlyA-3, Ting-1, T880, T710, T850) | Unidentified mutations | | 0.026 (6/226) | Unknown |

independently derived LOF alleles are all contiguous suggests that they may be beneficial or neutral in Central Europe.

*GS-OH*: The final major determinant of natural variation in Arabidopsis GSL chemotype is the GS-OH enzyme that adds a hydroxyl group to the carbon 2 on 3-butentyl GSL to create 2-OH-3-Butenyl GSL. Previous work had suggested two *GS-OH* alleles measurable in the seed, a functional allele in almost all accessions and a non-functional allele caused by active site mutations represented by the Cvi-0 accession (*Hansen et al., 2008*). Because of functional epistasis, we can only obtain functional phenotypic information from accessions that accumulate the GS-OH substrate, 3-Butenyl GLS. This identified 11 accessions with a non-functional GS-OH. Surveying these 11 accessions in the polymorph database (*The 1001 Genomes Consortium, 2016*) identified multiple independent LOF events. One of these 11 accessions has the Cvi active site mutations, two accessions have a shared nonsense SNP that introduces premature stop codons and two accessions have a complete loss of this gene (*Table 2*). The other six accessions with a loss of enzyme activity had an unidentified lesion due to sequence quality for this locus.

All these independent *GS-OH* LOF alleles are found in accessions that do not accumulate 3-Butenyl GSL, for example, three carbon or non-alkenyl accessions, suggesting that functional epistasis may be influencing the maintenance of these alleles in nature. Thus, we searched for the accessions that do not accumulate 3-Butenyl GLS and carry *GS-OH* LOF events (*Table 2*). In all cases, the LOF allele is more frequent in the non four carbon-alkenyl accessions than expected by random chance. This suggests that there is selection against 3-Butenyl GSL synthesis since LOF alleles are more frequent when the *GS-OH* gene is cryptic by functional epistasis. This agrees with the fact that the 3-Butenyl chemotype is the most sensitive to generalist lepidopteran herbivory (*Hansen et al., 2008*). Thus, these mutations may represent ongoing pseudogenization of the *GS-OH* gene when it is functionally hidden by epistasis at the *GS-AOP* and *GS-Elong* loci. These LOF events would then only be displayed upon rare admixture with 2-OH-3-Butenyl accessions.

## Discussion

Understanding the genetic, demographic and environmental factors that shape variation within a trait in a population is key to understanding trait evolution. In this work, we used a family of specialized metabolites, aliphatic GSLs, and measured their amounts in seeds of *A. thaliana* to query how genetics, geography, environment and demography intersect to shape chemotypic variation across Europe. We found that environmental conditions, together with geography, affect the presence and distribution of chemotypes within the accessions. This was demonstrated by specific traits that were associated with specific environmental conditions, and this association was shifted across the continent. Comparing the associations of traits to specific environmental conditions in Central Europe

versus the south revealed different behaviors. This demonstrated that chemotypic variation across Europe is created by a blend of all these processes that differ at the individual loci. This implies that a simultaneous analysis of both genotype and phenotype is required to fully interpret these processes and relationships. The above analysis is extensively using abiotic factors because of their availability while aliphatic GSLs have mainly been linked to biotic interactions. GSLs have been mainly linked to influencing biotic interaction with herbivores considered the primary drivers shaping GSL genetic diversity. However, because climate drives the distribution of biotic factors like herbivores, it is likely that climatic factors might appear as indirectly associated with GSLs. Interestingly, an aliphatic GSL was recently mechanistically linked to drought resistance in Arabidopsis, suggesting a potential role for abiotic factors to influence GSL diversity (*Salehin et al., 2019*). More work is needed to dissect all the potential components of an environment that influence selection on GSL chemotypes.

All three major aliphatic GSL loci display extensive allelic heterogeneity that is shaped by a blend of evolutionary events at each locus reminiscent of either parallel or convergent evolution. For this analogy, we are defining parallel evolution to be when a new chemotype arises two or more independent times by independent mutations from shared ancestral haplotype. Conversely, we are considering convergent evolution of a chemotype to occur when independent mutations in independent ancestral haplotypes derive the same chemotype. *GS-OH* provided clear evidence of parallel evolution where a single functional haplotype gave rise to at least four independent LOF alleles all with similar phenotypic consequence. *GS-AOP* suggested the potential for convergent evolution-style events leading to MSO chemotype arising from LOF events at the *GS-AOP* locus. The first characterized LOF event was in the Col-0 accession that has a 5 bp frameshift indel in the *AOP2* gene arising within the alkenyl *AOP2*-dominant *GS-AOP* haplotype. In this study, we identified additional parallel *AOP2* LOF events in this haplotype. More critical, we could identify multiple independent LOF events arising in the *AOP3* gene within the *AOP3*-dominant inversion *GS-AOP* haplotype. Thus, the same GSL chemotype, MSO, arises from independent LOF alleles in two different genes, *AOP2* and *AOP3*, that represent two different ancestral haplotypes (*Figure 6*). Thus, it appears that parallel-style events occur at all the loci and at least at the *GS-AOP* locus there is potential for a convergent-style evolutionary event leading to a single chemotype.

In addition to independent evolutionary events, three-way epistasis is shaping the allelic heterogeneity at these loci and their evolutionary potential. For example, the multiple independent *GS-OH* LOF variants all appear to have arisen in lineages where the *GS-AOP* and *GS-Elong* loci had haplotypes that epistatically combined to block the formation of the but-3-enyl GSL precursor for *GS-OH* (*Figure 2*). Thus, the parallel evolution of *GS-OH* LOF alleles is epistatically conditioned by *GS-AOP* and *GS-Elong*. A similar epistatic contingency also exists between the two independent *AOP3* alleles of *GS-AOP* (*Figure 6D and E*) and GS-Elong. Both of *AOP3* alleles of *GS-AOP* are coordinated with C3 haplotypes in GS-Elong. The *AZE* allele of *GS-AOP* is associated with a novel geographically limited *GS-Elong* allele in clade 8 (*Figure 5—figure supplement 5*) while the Central European *AOP3* allele is limited to accessions containing the clade 1, 3 or 6 C3 haplotypes of *GS-Elong*. There is also within-locus epistasis in the *GS-Elong* locus wherein a functional MAM1 leads to the creation of C4 GSLs regardless of the functional state of the *MAM2* gene and C3 haplotypes are marked by the loss of *MAM1*. All of these within- and between-loci interactions create a directional arrow for most loci where the haplotypes do not have equal evolutionary potential. For example, the clade 4 haplotype of *GS-Elong* can equally mutate to create a C3 or C4 chemotype because it has both *MAM1* and *MAM2*. However, the remaining clades like clade 3 have lost one or the other gene limiting their ability to create alternative chemotypes. In this case, the loss of *MAM1* likely prevents the ability for this C3 lineage to recreate the C4 chemotype. Thus, the potential evolutionary trajectory of a haplotype/allele at one or even within one of these GSL loci may be epistatically conditioned by the allelic state all the loci within a specific lineage.

This level of allelic diversity at these loci raises a question of how do pathways with this level of diversity and structural variation pass through speciation boundaries (*Figure 5—figure supplement 4*). Confounding this further is the observation that Brassica ssp. have genetic variation at *GS-Elong* and *GS-AOP* creating the exact same chemotypic variation found in Arabidopsis (*Heidel et al., 2006*; *Ramos-Onsins et al., 2004*; *Windsor et al., 2005*). There is similar within-species (*GS-AOP*) and between-species (*GS-AOP* and *GS-Elong*) variation between the closely related Arabidopsis sister species, *A. lyrata, A. petraea* and *A. halleri*. However, the underlying genetic basis is independent

events at the *AOP* and *MAM* loci showing that the variation did not go through the speciation boundary. Instead, this suggests that this variation has been recreated repeatedly in these species This raises the possibility that there may be a class of loci that are being repeatedly sampled by pangenomic variation across species within a family. To test this possibility would need a deeper phylogenetic sampling within and between species, particularly for understanding the intersection of ecology and evolution (*Göktay et al., 2021*; *Durvasula et al., 2017*).

Previous work on other biotic interactions genes like pathogen resistance gene-for-gene loci had indicated a predominant model of having two moderate-frequency ancient alleles creating the phenotypic variation within the species (*Atwell et al., 2010*; *Corrion and Day, 2001*; *MacQueen et al., 2016*). In contrast to R-gene loci characterized by old/stable biallelic variation, the GSL loci are characterized by a blend of structural and SNP-based variation with numerous alleles that appear young. In other cases, alleles of genes involved in biotic defense can present more complex patterns, for example, natural variation in the immune gene *ACCELERATED CELL DEATH 6* (*ACD6*) is caused by a rare allele causing an extreme lesion phenotype. It is not yet clear what selective pressures influence *ACD6* genetic variation (*Todesco et al., 2010*; *Zhu et al., 2018*). Thus, loci controlling resistance to diverse biotic traits under natural conditions have diverse genetic architectures and further work is needed to assess the range of allelic heterogeneity in these adaptive loci.

The allelic diversity at the GSL loci illustrates the benefit of simultaneously tracking the phenotype and genotype when working to understand the distribution of trait variation. For example, the Iberian Peninsula and the Mediterranean Basin had low variability in aliphatic GSL chemotypes, which show strong geographic structure. By contrast, Central/North Europe had high aliphatic GSL diversity with chemotypes showing overlapping geographic distributions. At first glance, this contrasts with previous work showing that the Iberian Peninsula and the Mediterranean Basin are genetically diverse. However, this discrepancy was caused by one of the causal loci. Specifically, the *GS-AOP* locus was largely fixed as the Alkenyl allele in Iberia/Mediterranean Basin with the alternative *GS-AOP* alleles enriched in Central Europe. In contrast to *GS-AOP*, Iberia and the Mediterranean Basin were highly genetically diverse for the *GS-Elong* locus and appear to contain almost all the variation in *GS-Elong* found throughout Europe (*The 1001 Genomes Consortium, 2016*). Thus, the chemotypic divergence from genomic variation expectations was driven by just the *GS-AOP* locus. This indicates that the high level of chemotypic variation in Central Europe is a blend of alleles that emerged in the south (*GS-Elong*) and alleles that possibly arose locally (*GS-AOP*, both nulls and *AOP3*). Further, the chemotypes found in any one region appear to be created by a combination of alleles as a result of a gene flow across the continent, local generation of new polymorphisms and local selective pressures.

Another challenge potentially caused by allelic heterogeneity and differential selective pressures, as displayed within this system, is detecting the known and validated causal natural variants within a population. Specifically, the GWA with this collection of 797 accessions was unable to find 80% of the known causal loci including one of the three major effect loci, *GS-OH*. Maximizing the number of genotypes and the SNP marker density was unable to overcome the complications imposed by the complex pressures shaping the distribution of these traits, potentially due to unequal dense sampling from the different areas. In this system, the optimal path to identifying the causal polymorphisms has instead been a small number of Recombinant Inbred Line populations derived from randomly chosen parents. In complex adaptive systems, the optimal solution to identifying causal variants is likely a blend of structured mapping populations and then translating the causal genes from this system to the GWA results and tracking the causal loci directly.

In this work, we combined different approaches to uncover some of the parameters shaping the aliphatic GSL content across Europe. Widening the size of the population will enable us to deepen our understanding on the evolutionary mechanisms shaping a phenotype in a population.

## Materials and methods
### Plant material
Seeds for 1135 Arabidopsis (*A. thaliana*) genotypes were obtained from the 1001 genomes catalog of *A. thaliana* genetic variation (https://1001genomes.org/). All Arabidopsis genotypes were grown at 22°C/24°C (day/night) under long-day conditions (16 hr of light/8 hr of dark). Two independent replicates were performed, each of them included the full set of genotypes. The replicates obtained

from independent maternal plants were grown in randomized fashion. In the analyses, only accessions from Europe and around Europe were included (*Figure 3A*), resulting in an analysis of 797 accessions. A list of the accessions can be found in *Supplementary file 1*.

## GSL extractions and analyses

GSLs were measured as previously described (*Kliebenstein et al., 2001a*; *Kliebenstein et al., 2001b*; *Kliebenstein et al., 2001c*). Briefly, ~3 mg of seeds were harvested in 200 μL of 90% methanol. Samples were homogenized for 3 min in a paint shaker, centrifuged, and the supernatants were transferred to a 96-well filter plate with DEAE sephadex. The filter plate with DEAE sephadex was washed with water, 90% methanol and water again. The sephadex-bound GSLs were eluted after an overnight incubation with 110 μL of sulfatase. Individual desulfo-GSLs within each sample were separated and detected by HPLC-DAD, identified, quantified by comparison to standard curves from purified compounds and further normalized to the weight. A list of GSLs and their structure is given in *Supplementary file 1A*. Raw GSLs data are given in *Supplementary file 1B*.

## Statistics, heritability and data visualization

Statistical analyses were conducted using R software (https://www.R-project.org/) with the RStudio interface (http://www.rstudio.com/). For each independent GLS, a linear model followed by ANOVA was utilized to analyze the effect of accession, replicate and location in the experiment plate upon the measured GLS amount. Broad-sense heritability (*Supplementary file 1C*) for the different metabolites was estimated from this model by taking the variance due to accession and dividing it by the total variance. Estimated marginal means (emmeans) for each accession were calculated for each metabolite from the same model using the package emmeans (*CRAN, 2021a*; *Supplementary file 1D*). PCAs were done with FactoMineR and factoextra packages (*Abdi and Williams, 2010*). Data analyses and visualization were done using R software with tidyverse (*Wickham et al., 2019*) and ggplot2 (*Kahle and Wickham, 2013*) packages.

Maps were generated using ggmap package (*Kahle and Wickham, 2013*).

## Phenotypic classification based on GSL content

For each accession, the expressed enzyme in each of the following families was determined based on the content (presence and amounts) of short-chained aliphatic GSLs.

*MAM enzymes*: The total amount of three carbon GSLs and four carbon GSLs was calculated for each accession. Three carbon GSLs include 3MT, 3MSO, 3OHP and Allyl GSL. Four carbon GSLs include 4MT, 4MSO, 4OHB, 3-Butenyl and 2-OH-3-Butenyl GSL (for structures and details, see *Supplementary file 1*). Accessions that the majority of aliphatic short-chained GSL contained three carbons in their side chains were classified as *MAM2* expressed (*Figure 4—figure supplement 1*). Accessions that the majority of aliphatic short-chained GSL contained four carbons in their side chains were classified as *MAM1* expressed (*Figure 4—figure supplement 1*). The accessions were plotted on a map based on their original collection sites (*Figure 4—figure supplement 1*).

*AOP enzymes*: The relative amount of alkenyl GSL, alkyl GSL and MSO GSL was calculated in respect to the total short-chained aliphatic GSL as follows:

$$Alkenyl\,GSL\,(AOP2\;expressed) = \frac{\text{Allyl} +2-\text{OH}-3-\text{butenyl} + 3-\text{butenyl}}{\text{Total short chained GSL}}$$

$$Alkyl\,GSL\,(AOP3\;expressed) = \frac{\text{3OHP} + \text{4OHB}}{\text{Total short chained GSL}}$$

$$MSO\,GSL\,(AOP\;null) = \frac{\text{3MSO} + \text{4MSO}}{\text{Total short chained GSL}}$$

The expressed AOP enzyme was determined based on those ratios: Accessions with majority alkenyl GSL were classified as *AOP2* expressed. Accessions with majority of alkyl GSL were classified as *AOP3* expressed. Accessions with majority of MSO GSL were classified as *AOP* null. The accessions were plotted on a map based on their original collection sites (*Figure 4—figure supplement 2*).

*GS-OH enzyme*: The ratio between 2-OH-3-Butenyl GSL to 3-Butenyl GSL was calculated only for *MAM1*-expressed accessions (accessions that the majority of GSLs contain four carbons in their side chain). Accessions with high amounts of 2-OH-3-Butenyl GSL were classified as *GS-OH* functional. Accessions with high amounts of 3-Butenyl GSL were classified as *GS-OH* non-functional. The accessions were plotted on a map based on their original collection sites (*Figure 4—figure supplement 3*).

Each accession was classified to one of seven aliphatic short-chained GSLs based on the combination of the dominancy of the enzymes as follows: MAM2, AOP null: classified as 3MSO dominant. MAM1, AOP null: classified as 4MSO dominant. MAM2, AOP3: classified as 3OHP dominant. MAM1, AOP3: classified as 4OHB dominant. MAM2, AOP2: classified as Allyl dominant. MAM1, AOP2, GS-OH non-functional: classified as 3-Butenyl dominant. MAM1, AOP2, GS-OH functional: classified as 2-OH-3-Butenyl dominant. The accessions were plotted on a map based on their original collection sites and colored based on their dominant chemotype (*Figure 4*).

## Environmental and demographic data

Environmental and demographic data (referred to as 'genomic group') were obtained from the 1001 genomes website (https://1001genomes.org/, for geographical and demographic data) and from the Arabidopsis CLIMtools (http://www.personal.psu.edu/sma3/CLIMtools.html, *Ferrero-Serrano and Assmann, 2019*) for environmental data. We chose the five variables that captured a majority of the variance in this dataset based on PCA using different combinations of variables. The chosen variables are maximal temperature of warmest month (WC2_BIO5), minimal temperature of coldest month (WC2_BIO6), precipitation of wettest month (WC2_BIO13), precipitation of driest month (WC2_BIO14) and distance to the coast (in km). Each one of the above variables (including genomic group) was assigned to each one of the accessions.

## Environmental models

Linear models to test the effect of geographical and environmental parameters (*Figure 3—figure supplement 1* and *Figure 4—source data 1*) were conducted using dplyr package (*CRAN, 2021b*) and included the following parameters:

*Figure 3—figure supplement 1* linear models for collection sites: PC score ~ Latitude + Longitude + Latitude * Longitude.

*Table 1* and *Figure 4—source data 1* for all the data: C length (C3 and C4) or the chemotypes (Allyl and 2-OH-3Butenyl) ~ Genomic group + Geography (north versus south) + Max temperature of warmest month + Min temperature of coldest month + Precipitation of wettest month + Precipitation of driest month + Distance to the coast + Geography * Genomic group + Geography * Max temperature of warmest month + Geography * Min temperature of coldest month + Geography * Precipitation of driest month + Geography * Precipitation of wettest month + Geography * Distance to the coast.

For the north and the south: C length (C3 and C4) or the chemotypes (Allyl and 2-OH-3Butenyl) ~ Genomic group + Geography (north versus south)+ Max temperature of warmest month + Min temperature of coldest month + Precipitation of wettest month + Precipitation of driest month + Distance to the coast.

## Genome-wide association studies

The phenotypes for GWA studies were each accession value for PC1 and 2. GWA was implemented with the easyGWAS tool (*Grimm et al., 2017*) using the EMMAX algorithms (*Kang et al., 2010*) and a minor allele frequency (MAF) cutoff of 5%. The results were visualized as Manhattan plots using the qqman package in R (*Turner, 2014*).

## Phylogeny

Genomic sequences from the accessions for *MAM3* – AT5G23020, *AOP2* – Chr4, 1351568 until 1354216, *AOP3* – AT4G03050.2, *GS-OH* – AT2G25450 and *MYB37* – AT5G23000 were obtained using the Pseudogenomes tool (https://tools.1001genomes.org/pseudogenomes/#select_strains).

Multiple sequence alignment was done with the msa package (default settings) in R using the ClustalW, ClustalOmega and Muscle algorithms (*Bodenhofer et al., 2015*). Phylogenetic trees were generated with the 'ape' package (neighbor-joining tree) (*Paradis and Schliep, 2019*) and were visualized with ggtree package in R (*Yu, 2020*). Each tree was rooted by the genes matching *A. lyrata's* functional orthologue or closest homologue.

Bootstrap analyses (Bootstrap = 100) was done with 'ape' package in R (*Paradis and Schliep, 2019*), with the same tree inference method as described before. For *MAM3* bootstrap analysis, the accessions with low-quality sequencing were excluded.

*Amino acid phylogenies*: Sequences were taken from *Abrahams et al., 2020*, which uses *A. thaliana* Col-0 genome and the *MAM2* amino acid sequence 1006452109 from the Arabidopsis Information Resource (TAIR) database. Alignments were run using MAFFT (*Katoh et al., 2017*; *Kuraku et al., 2013*) and cleaned using Phyutility at a 50% occupancy threshold (*Smith and Dunn, 2008*). RAxML was used for phylogenetic inference (*Stamatakis, 2014*) with the PROTCATWAG model (Bootstrap = 1000).

## Sequencing

PacBio long read-based de novo genome assemblies of the relevant accession were generated as part of the 1001 Genomes Plus project. The genomes were assembled with Canu (v1.71) (*Koren et al., 2017*) and polished using the long reads followed by a second polishing step with PCR-free short reads.

## Acknowledgements

This work was supported by the National Science Foundation, Directorate for Biological Sciences, Division of Molecular and Cellular Biosciences (grant no. MCB 1906486 to DJK) and Division of Integrative Organismal Systems (grant no. IOS 1655810 to DJK, and IOS 1754201 to RA), and by the United States-Israel Binational Agricultural Research and Development Fund (to DJK and EK, grant no. FI-560-2017). We thank Dr. Allison Gaudinier (Department of Plant and Microbial Biology, University of California, Berkeley), Dr. Tobias Züst (Institute of Plant Sciences, University of Bern), Dr. Christopher W Wheat (Department of Zoology, Stockholm University) and Dr. Daniel Runcie (Department of Plant Sciences, University of California Davis) for critical reading of the manuscript. We thank the 1001 Genomes Plus project for access for newly assembled *A. thaliana* accession genomes. The 1001 Genomes Plus project is funded by ERA-CAPS through BBSRC, DFG and FWF to Paul Kersey, Detlef Weigel and Magnus Nordborg, respectively.

## Additional information

### Competing interests

Daniel J Kliebenstein: Reviewing editor, eLife. The other authors declare that no competing interests exist.

### Funding

| Funder | Grant reference number | Author |
| --- | --- | --- |
| National Science Foundation | MCB 1906486 | Daniel J Kliebenstein |
| United States - Israel Binational Agricultural Research and Development Fund | FI-560-2017 | Ella Katz<br>Daniel J Kliebenstein |
| National Science Foundation | IOS 1655810 | Daniel J Kliebenstein |
| National Science Foundation | IOS 1754201 | Ruthie Angelovici |

The funders had no role in study design, data collection and interpretation, or the decision to submit the work for publication.

### Author contributions

Ella Katz, Conceptualization, Data curation, Formal analysis, Visualization, Writing – original draft; Jia-Jie Li, Benjamin Jaegle, Shawn R Abrahams, Clement Bagaza, Samuel Holden, Data curation; Haim Ashkenazy, Data curation, Writing – review and editing; Chris J Pires, Ruthie Angelovici, Writing – review and editing; Daniel J Kliebenstein, Conceptualization, Data curation, Formal analysis, Funding acquisition, Visualization, Writing – original draft

## Author ORCIDs

Ella Katz ⓘ http://orcid.org/0000-0003-1619-5597
Haim Ashkenazy ⓘ http://orcid.org/0000-0002-5079-4684
Shawn R Abrahams ⓘ http://orcid.org/0000-0003-1749-2040
Daniel J Kliebenstein ⓘ http://orcid.org/0000-0001-5759-3175

## Decision letter and Author response

Decision letter https://doi.org/10.7554/eLife.67784.sa1
Author response https://doi.org/10.7554/eLife.67784.sa2

## Additional files

### Supplementary files

• Supplementary file 1. GSLs data. (A) List of GSLs and structures. (B) Accessions and glucosinolate (GSL) data – raw data. (C) Heritability values. (D) Accessions and GSL data – emmeans.

• Supplementary file 2. SNPs in glucosinolates (GSLs)-related loci under different genome-wide association (GWA) studies: GSL values were used as traits to conduct GWA studies. The number of significant SNPs in the GSLs related loci (columns c to o) was counted for each GWA study separately. Rows 2–3: common name and AT number of gene/s in the loci. Rows 4–5: upstream and downstream positions of the relevant loci (10 kb were added upstream and downstream of the genes). Rows 6–33: GSLs traits used for GWA studies. In black: number of SNPs with p value between 0.00001 and 0.0000001. In red: number of SNPs with p value equal or smaller than 0.0000001.

• Transparent reporting form

### Data availability

All data generated or analyzed during this study are included in the manuscript and supporting files.

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
