## [Decision Letter]

**Acceptance summary:**

The manuscript by Katz and colleagues makes key advances in understanding how glucosinolate chemotype variation across Arabidopsis ecotypes is linked to the underlying genetic variation. It demonstrates the complexity of the interplay between genetic and environmental factors and gives insights into how complex traits could evolve in natural populations. This article reveals the complexities involved in conducting genome-wide association studies on geographically structured populations which cannot be eliminated even in a very well studied system. It thus bears important general lessons for researchers in the genomic era.

**Decision letter after peer review:**

[Editors’ note: the authors submitted for reconsideration following the decision after peer review. What follows is the decision letter after the first round of review.]

Thank you for submitting your work entitled "Genetic variation, environment and demography intersect to shape Arabidopsis defense metabolite variation across Europe" for consideration by *eLife*. Your article has been reviewed by 3 peer reviewers, and the evaluation has been overseen by a Reviewing Editor and a Senior Editor. The following individuals involved in review of your submission have agreed to reveal their identity: Arthur Korte (Reviewer #1); Juergen Kroymann (Reviewer #3).

Our decision has been reached after consultation between the reviewers. Based on these discussions and the individual reviews below, we regret to inform you that your work will not be considered further for publication in *eLife*. However, the reviewers and editors agree that both the topic and the general approach are of interest. *eLife* would consider a substantially revised version of the manuscript which addresses the points raised by the individual reviewers and summarized below. Because we expect that these revisions represent substantial work, we would consider a revised manuscript as a new submission.

Specifically, the reviewers and editors agree that the study could provide an important new insight into patterns and processes of evolution. In the words of Reviewer 1, it is valuable "to present the complexity of the interplay between genetic and environmental factors and its contribution to trait evolution in natural populations." It is perhaps especially valuable to do so in such a well-studied system as *Arabidopsis thaliana*, as this indicates fundamental complexities which cannot be remedied by large volumes of data alone, but require well-informed approaches to analyze the data and in some cases, perhaps, cannot be resolved. Thus although we are not convinced that any one aspect of this study is very novel, we do think that the study as a whole presents an unusually holistic perspective and thus could be a very valuable contribution. Nevertheless, we also agree that there are substantial gaps and pitfalls in the analyses as they are currently presented, and these must be remedied in order for the conclusions of the study to be well supported:

1. We are concerned that MAM3 is not a good proxy for the GS-Elong locus, and there is evidence of recombination within MAM3; and that resequencing of this locus in the accessions studied is required to support the conclusions that the authors wish to draw in this study. Please see detailed comments from Reviewer 3.

2. Several sections of the study require deep revision in terms of writing and analysis. See major comments from Reviewer 2, as well as the other reviewers.

*Reviewer #1:*

The authors present a detailed and well-written analyses of the variation in the glucosinolate chemotype of different Arabidopsis ecotypes. These differences were linked to the underlying genetic variation. They could show many complex patterns that are related to the distinct chemotypes. The analyses highlights the complexity of the interplay between genetic and environmental factors and gives insights into how traits evolve in natural populations.

I only have one point that I was wondering about.

23 different GSL have been detected and quantified and GWAS has been performed on PCs that summarize the GSL variation. The GWAs on PC1 and PC2 do only identify the two known loci AOP and NAM and no additional loci are significant. As this first two PCs explain only 33 % of the total phenotypic variation, I wonder if GWAS on the individual GSL concentration, especially as all of these show a high heritability, could identify additional associations? It would be nice to have an overview of these potential associations.

*Reviewer #2:*

This manuscript describes the contribution of environment and demography to shape the genetic variation underlying the glucosinolate (GLS) composition of *Arabidopsis thaliana* across Eurasia. To this end, the authors carried out a very extensive characterization of different GLSs in a large collection of natural Arabidopsis accessions. Genetic analyses of these data show that the epistasis underlying causal loci hampers the identification of new loci. More importantly, authors compare these data with environmental factors to show significant environmental associations, which support a differential adaptive role of this variation across Europe. In addition, thorough genetic analyses of the major underlying genes identified multiple independent mutations in several duplicated genes, which demonstrate a repeated evolution pattern with differential geographic histories. Overall, this study shows a complex adaptive history of Arabidopsis plant defenses mediated by GLSs at a continental scale and brings forward our current understanding of adaptation of plant defense mechanisms.

Although most sections of the manuscript are written in a comprehensive manner, several sections of results and discussion are rather confusing due to lack of information on particular analyses. The following sections should be revised:

1. The section entitled "Geography and environmental parameters affect GLS variation" (pages 13-14) is very confusing due to an unclear distinction between demography and environment. Given the strong geographic structure of Arabidopsis genetic diversity across Europe (The 1001 Genomes Consortium, 2016), most genetic diversity can be expected to correlate with environment due to Arabidopsis demographic history but not reflecting true adaptive association. For this reason, to distinguish true environmental associations from correlations related with demographic history, analyses must take into account the genetic structure (genetic or genomic relationships among accessions). Usually, the effect of demographic factors on the distribution of genetic diversity is inferred by the genetic structure. According to Supp. Figure 7 and Materials and Methods, authors seem to have included such genetic structure in their MANOVA analyses, which is referred to as genomic group. However, authors should describe what are those genomic groups, and if they were included also in Figure 4B. As shown by Supp. Figures 7A and 8, the genetic structure explains most of the diversity in MANOVA and Random forest models, indicating that most of the geographic patterns can be explained by the demographic history. However, both Figures also show that there is still a significant association with environmental factors, which supports that the environment contributes to maintain the genetic diversity of chemotypes. The text describing these results should be carefully rewritten to explain how the genetic structure (demography) has been taken into account and separating clearly demographic and environmental components. The genetic structure is only mentioned in line 259 where it is confusingly named as "ancestral state".

2. The section describing the genetic diversity of MAM3 across Europe (lines 302-326) shows certain overinterpretation of the action of natural selection, which should be revised and rewritten in a softer tone. For instance, the sentence in line 306 stating "If the latter is true, this would argue for a selective pressure shaping this C3/C4 divide". The two scenarios described in paragraph of lines 302-306 could be explained by natural selection or demography, and one should be cautious interpreting geographic patterns. In particular, the geographic pattern observed in Iberia is just showing that this region contains a large diversity for C3/C4 chemotypes and MAM3, as shown previously for most genes in this region (The 1001 Genomes Consortium). Therefore, the sentence in lines 310-313 ("This suggests that the strong geographic partitioning of the C3/C4 chemotypes in Iberia may be driven by selective pressure causing the partitioning of the chemotypes rather than neutral demographic processes") or lines 324-326 ("…,Iberia, showing evidence of local selection while other regions, central Europe, possibly showing a blend requiring further work to delineate") are not fully supported by data. In addition, this section on MAM3 could include the recent study on the genetic diversity of Europe, which suggests that Iberia could have also been colonized from central Europe after the last glaciation (Lee et al., 2017; Nature Communications 8:14458).

3. The section describing GS-OH gene and Table 1 (lines 377-390) is rather confuse. In line 377 is stated: "We could not identify the causal LOF in the other six accessions due to sequence quality". However, Table 1 shows the structure of GS-OH gene with apparent location of six mutations. It is unclear which are those six accessions of Table1, what is represented in the structure of the genes, and the meaning of symbols on those genes (A and the vertical bars). Are those accessions of Table 1 the same described in the text? It should also made clear the type of mutations of those six accessions instead of the confusing term "Independent mutation". Furthermore, this section refers to Supplementary Table 2 (line 383), but I could not find such table.

*Reviewer #3:*

The manuscript is based on an impressive data set of (seed) glucosinolate profiles across Europe and beyond, with a focus on methionine-derived glucosinolates. However, the manuscript reads in many aspects like a 'best of' of previous research, with limited net gain in insight. In addition, there are substantial weaknesses (in particular relating to GS-Elong) that prevent me from recommending this manuscript for publication in *eLife* its current state.

1. The representation of the MAM reaction is misleading and incorrect (lines 101ff). MAMs catalyze condensation of 2-oxoacids with acetylCoA. Two further reaction steps, catalyzed by an isomerase and a dehydrogenase, are necessary to complete chain extension. This extension consists formally in the addition of a methylene group per reaction cycle. With MAM2, the net result is an extension by one (not two!) methylene group, with MAM1 it is mostly two; note that methionine has already two methylene groups in the side chain! I suggest replacing the reference Abrahams et al., 2020, with Graser et al., 2000 (Archives of Biochemistry and Biophysics, 378, 411-419). Furthermore, Kroymann et al., 2001 (Plant Phys. 127, 1077-1088), Benderoth et al., 2006 and Textor et al., 2007 (Plant Phys. 144, 60-71) should be added as original references for the MAM1, 2 and 3 reactions.

2. The use of MAM3 as a proxy for inferring the configuration of the GS-Elong locus is highly debatable. Even though recombination appears to be suppressed at and around GS-Elong, an additional gene (or additional genes) upstream of GS-Elong should be used to construct (a) gene tree(s) and compare with the MAM3 gene tree for congruency. In any case, without sequencing of the MAM genes at the GS-Elong locus, it is not possible to conclude that 'local' configurations of MAM1/MAM2 have not arisen in Iberia (or elsewhere).

3. The *A. lyrata* ortholog of Arabidopsis MAM3 is MAMb (not MAMc). The MAM3 gene tree does not provide any statistical support for the branching order. Therefore, it remains unclear whether the presented tree reflects the true branching order and whether the 'most basal' clade in the tree is indeed the most basal. Note also that the Cal-0 gene at the MAM2 position is actually mostly MAM2, except for a converted stretch towards the 3' end of the gene; see: Figure 2 in Kroymann et al., 2003. Hence, the most parsimonious interpretation of available data is that the ancestral state of GS-Elong in Arabidopsis is indeed: MAM2, MAM1 and MAM3. Note also that the Arabidopsis genes in Abrahams et al., 2020, are from TAIR 10, i.e. from Col-0 (see Table 1 in Abrahams et al., 2020). Because Col-0 does not have MAM2, this gene is not included in the gene trees presented by Abrahams et al., 2020.

4. Despite the use of an impressive number of accessions, sufficiently dense sampling appears to be only available for the Iberian peninsula, the southern coast of Sweden and the south-western coast of Italy. In this regard, the authors overstate their findings (e.g., lines 219ff).

5. While the link between glucosinolates and biotic stress is well documented, evidence for a major role of abiotic factors in shaping glucosinolate profiles is less well substantiated. Now, the authors seem to find that certain chemotypes are favored by high precipitation in 'the North' but that the same chemotypes are favored by low precipitation in 'the South'. This is interesting, but what could explain this unexpected pattern? Any idea? Furthermore, with additional sampling on the eastern coast of Italy and the eastern coast of the Adriatic, would you expect to obtain similar patterns?

6. Please, make sure that the citation style is uniform. Please also make sure that the tense is uniform throughout the manuscript. Please, simplify the introduction in a way that it is accessible by non-specialists. Please improve Figure 5D – this is a mess.

[Editors’ note: further revisions were suggested prior to acceptance, as described below.]

Thank you for submitting your article "Genetic variation, environment and demography intersect to shape Arabidopsis defense metabolite variation across Europe" for consideration by *eLife*. Your article has been reviewed by 2 peer reviewers, and the evaluation has been overseen by a Reviewing Editor and Meredith Schuman as the Senior Editor. The following individual involved in review of your submission has agreed to reveal their identity: Arthur Korte (Reviewer #2).

Essential Revisions:

While we find the overall story to be insightful and advancing the field, we do think that some of the analyses still need to be redone, while the eco-evolutionary framework needs to be better defined.

1. We have serious concerns regarding the environmental association analyses. Since there is a general correlation between genetic/genomic/phenotypic diversity and environmental factors due to Arabidopsis demographic history, the overall genetic structure needs to be included in the statistical models testing the associations between traits and environmental factors. You should produce tables similar to that of Supplementary Figure 8A for the chemotypes of Figure 4 and derive statistical significances from these tables, in order to provide convincing evidence of the differential climatic patterns in both European regions.

2. The conceptual framework of repeated evolution that differentiates between parallel and convergent evolution was traditionally developed for interspecific evolution, not for intraspecific evolution where it refers to parallel evolution by multiple independent mutations in the same gene. The differentiation between parallel and convergent evolution for intraspecific evolution is not so straightforward. Therefore we want you to provide a more precise definition of intraspecific convergent and parallel chemotype evolution and outline the role of different or the same genes from a tandem repeat cluster, as well as the role of epistasis between duplicated genes, therein. To this end, the LD analysis between MAM genes and flowering genes should also be removed from the argumentation, because this analysis clearly shows that LD between these genes is not larger than average LD between random pairs of genes. Therefore, the conclusion drawn from this LD analysis is not supported.

We are optimistic that regardless of the outcome of the corrected environmental analysis (point 1), this study represents a valuable contribution, especially if the revised framework for intraspecific repeated evolution – helping to conceptually unite studies of population and functional genetics in an apt system – can be articulated and illustrated more clearly in the revised analysis (point 2).

*Reviewer #1 (Recommendations for the authors):*

In this new version of the Manuscript by Katz et al., authors address numerous important questions raised in the previous version, and rewrite many of the paragraphs that were confusing or incorrect. In this version authors provide better support for some of their conclusions in a more comprehensive manuscript. However, this Reviewer still has serious concerns regarding the environmental association analyses carried out in this study to support their conclusions. In addition, for clarity and comprehension I also suggest addressing an important conceptual issue, as well as the numerous editorial points described below.

1A. As explained in previous Reviewer 2 report to Authors, in *A. thaliana* there is a general correlation between genetic/genomic/phenotypic diversity and environmental factors due to Arabidopsis demographic history. For this reason, the overall genetic structure needs to be included in the statistical models testing the associations between traits and environmental factors. Authors now provide more detailed description of the statistical tests applied for such associations. However, as stated in Materials and methods, the statistical models used to build Figure 4 and Supplemental Figure 8B do not include the genomic groups. Therefore, significant associations described in these Figures and throughout the manuscript are mainly reflecting the association of genomic groups (demographic history) with climate. These two figures are the main support for the differential environmental associations in Northern and Southern Europe, but they just show that different genomic groups are present in each geographic area. It seems that such differential association between climate and chemotypes might be true, as shown by the table of Supplementary Figure 8A, where authors include the genomic groups to test the differential association specifically for C3 and C4 status. However, in this table the variable "Precipitation for the driest moth" appears as non-significantly associated. In spite of this, authors choose this variable in Supplementary Figure 8B (as well as in Figure 4A) to illustrate that this climatic variable is associated when applying a simple t-test to compare both groups. This test does not support such environmental association because it is not controlling for the genomic group (demographic history). By contrast, the table of Supplementary Figure 8 shows a stronger differential association in Northern and Southern Europe for minimum temperature, which is associated only in the south but not in the north. Tables similar to that of Supplementary Figure 8A should be provided for the chemotypes of Figure 4, and statistical significances should be derived from them, in order to provide convincing evidence of their differential climatic patterns in both European regions.

1B. Related with the environmental associations, authors include random forest analyses. However, in the way these analyses are presented, they are hard to interpret because variables with the lowest mean decrease Gini (the probability of the variable of being wrong according to Authors) mostly have the lowest mean decrease accuracy (suitability of the variable as predictor).

2. A major goal of this manuscript is the distinction between convergent and parallel evolutions in GSL chemotypes, which are described as two different patterns of repeated evolution. However, these concepts have been classically developed to characterize interspecific evolution, and their translation into the intraspecific variation is not straight forward, thus requiring a precise definition of both alternatives. Authors define parallel and convergent evolution as traits derived from either the same or different "genetic backgrounds". In this definition, the concept of genetic background is very vague and it is unclear what is meant. Authors apply these two concepts to the particular case of repeated evolution of GSL chemotypes evolved specifically from two tandem duplicated genes (MAM and AOP loci). Given the epistasis between these pairs of duplicated genes, the same trait can be derived from the same or different gene within. Then, it seems to this Reviewer that Authors mean by parallel and convergent evolution, the evolution occurring from the same or different gene of the cluster, respectively. In my opinion, this dissection of repeated evolution for the specific case of duplicated genes with epistasis is a major contribution of this study, but it is hard to understand. The manuscript and the audience would benefit by making these concepts clearer. I suggest that Authors add this in discussion, (e.g. paragraph of lines 550-554), explaining the role of different or the same gene from tandem repeat cluster, as well as the role of epistasis between duplicated genes for the distinction of their concepts of parallel and convergent evolution.

*Reviewer #2 (Recommendations for the authors):*

The authors present a substantial revised version of their manuscript that improved in clarity. All my concerns and suggestions are addressed in this version and I don't have any further comments.

---

## [Author Response]

[Editors’ note: the authors resubmitted a revised version of the paper for consideration. What follows is the authors’ response to the first round of review.]

1. We are concerned that MAM3 is not a good proxy for the GS-Elong locus, and there is evidence of recombination within MAM3; and that resequencing of this locus in the accessions studied is required to support the conclusions that the authors wish to draw in this study. Please see detailed comments from Reviewer 3.2. Several sections of the study require deep revision in terms of writing and analysis. See comment from Reviewer 2, as well as the other reviewers.

We have worked extensively to address both of these concerns in the revised manuscript. There are more details below about the specifics. In summary, we have worked on the entire manuscript to work on the wording and clarification in all noted sections and in the entire manuscript. Specifically, we have tried to make the holistic aspect of the study clearer. For the MAM locus, we have redone the analysis with the flanking gene to test for within locus recombination and how this may influence the analysis. Further, we have obtained 11 new sequences for this locus to further test the use of MAM3 as a proxy. Both approaches support the use of the MAM3 as a proxy to derive general conclusions. Finally, we have reworked all discussion on this locus, the enzymes chemistry and its potential evolution to reflect what is and is not supported by the phylogenetics.

Reviewer #1:The authors present a detailed and well-written analyses of the variation in the glucosinolate chemotype of different Arabidopsis ecotypes. These differences were linked to the underlying genetic variation. They could show many complex patterns that are related to the distinct chemotypes. The analyses highlights the complexity of the interplay between genetic and environmental factors and gives insights into how traits evolve in natural populations.I only have one point that I was wondering about.23 different GSL have been detected and quantified and GWAS has been performed on PCs that summarize the GSL variation. The GWAs on PC1 and PC2 do only identify the two known loci AOP and NAM and no additional loci are significant. As this first two PCs explain only 33 % of the total phenotypic variation, I wonder if GWAS on the individual GSL concentration, especially as all of these show a high heritability, could identify additional associations? It would be nice to have an overview of these potential associations.

To test the reviewer’s hypothesis, we have performed GWAS using the accumulation of each of the 23 individual GSL to test if this identifies more associations both with known GSL genes and unknown genes. This analysis did find a few additional SNPs in unknown genes, it failed to detect additional GSL related genes similar to the use of principal components. We added a supplemental figure of the 23 Manhattan plots showing it, and a supplementary table that summarize the number of SNPs in each of the expected GSL genes (Figure 3 - figure supplement 3 and Supplementary file 2).

Reviewer #2:This manuscript describes the contribution of environment and demography to shape the genetic variation underlying the glucosinolate (GLS) composition of *Arabidopsis thaliana* across Eurasia. To this end, the authors carried out a very extensive characterization of different GLSs in a large collection of natural Arabidopsis accessions. Genetic analyses of these data show that the epistasis underlying causal loci hampers the identification of new loci. More importantly, authors compare these data with environmental factors to show significant environmental associations, which support a differential adaptive role of this variation across Europe. In addition, thorough genetic analyses of the major underlying genes identified multiple independent mutations in several duplicated genes, which demonstrate a repeated evolution pattern with differential geographic histories. Overall, this study shows a complex adaptive history of Arabidopsis plant defenses mediated by GLSs at a continental scale and brings forward our current understanding of adaptation of plant defense mechanisms.Although most sections of the manuscript are written in a comprehensive manner, several sections of results and discussion are rather confusing due to lack of information on particular analyses. The following sections should be revised:1. The section entitled "Geography and environmental parameters affect GLS variation" (pages 13-14) is very confusing due to an unclear distinction between demography and environment. Given the strong geographic structure of Arabidopsis genetic diversity across Europe (The 1001 Genomes Consortium, 2016), most genetic diversity can be expected to correlate with environment due to Arabidopsis demographic history but not reflecting true adaptive association. For this reason, to distinguish true environmental associations from correlations related with demographic history, analyses must take into account the genetic structure (genetic or genomic relationships among accessions). Usually, the effect of demographic factors on the distribution of genetic diversity is inferred by the genetic structure. According to Supp. Figure 7 and Materials and methods, authors seem to have included such genetic structure in their MANOVA analyses, which is referred to as genomic group. However, authors should describe what are those genomic groups, and if they were included also in Figure 4B. As shown by Supp. Figures 7A and 8, the genetic structure explains most of the diversity in MANOVA and Random forest models, indicating that most of the geographic patterns can be explained by the demographic history. However, both Figures also show that there is still a significant association with environmental factors, which supports that the environment contributes to maintain the genetic diversity of chemotypes. The text describing these results should be carefully rewritten to explain how the genetic structure (demography) has been taken into account and separating clearly demographic and environmental components. The genetic structure is only mentioned in line 259 where it is confusingly named as "ancestral state".

We apologize for not making it clearer that we had included population structure in these analyses and we have worked to clarify this and to ensure that it is clear for each analysis how population structure/demography was included. For the MANOVA and Random Forest models, population structure was included using the genomic groupings as previously defined in The 1001 Genomes Consortium, 2016.

2. The section describing the genetic diversity of MAM3 across Europe (lines 302-326) shows certain overinterpretation of the action of natural selection, which should be revised and rewritten in a softer tone. For instance, the sentence in line 306 stating "If the latter is true, this would argue for a selective pressure shaping this C3/C4 divide". The two scenarios described in paragraph of lines 302-306 could be explained by natural selection or demography, and one should be cautious interpreting geographic patterns. In particular, the geographic pattern observed in Iberia is just showing that this region contains a large diversity for C3/C4 chemotypes and MAM3, as shown previously for most genes in this region (The 1001 Genomes Consortium). Therefore, the sentence in lines 310-313 ("This suggests that the strong geographic partitioning of the C3/C4 chemotypes in Iberia may be driven by selective pressure causing the partitioning of the chemotypes rather than neutral demographic processes") or lines 324-326 ("…,Iberia, showing evidence of local selection while other regions, central Europe, possibly showing a blend requiring further work to delineate") are not fully supported by data. In addition, this section on MAM3 could include the recent study on the genetic diversity of Europe, which suggests that Iberia could have also been colonized from central Europe after the last glaciation (Lee et al., 2017; Nature Communications 8:14458).

We rewrote this section, to both soften and better explain our hypothesis. We emphasized the fact that though there is high diversity in the haplotypes across Iberia, there is a clean delineation of the two opposing chemotypes controlled by these haplotypes. Further, the haplotypes within a chemotype are randomly distributed across the portions of Iberia containing that specific chemotype. This suggests that the haplotypes within a chemotype are more random across the landscape than the chemotype providing evidence supportive of selective delineation but not definitive. We also added the necessary citation.

3. The section describing GS-OH gene and Table 1 (lines 377-390) is rather confuse. In line 377 is stated: "We could not identify the causal LOF in the other six accessions due to sequence quality". However, Table 1 shows the structure of GS-OH gene with apparent location of six mutations. It is unclear which are those six accessions of Table1, what is represented in the structure of the genes, and the meaning of symbols on those genes (A and the vertical bars). Are those accessions of Table 1 the same described in the text? It should also made clear the type of mutations of those six accessions instead of the confusing term "Independent mutation". Furthermore, this section refers to Supplementary Table 2 (line 383), but I could not find such table.

We prepared a new version for table 1. In this table we presented a clearer description of what the mutations are (and which ones were not identifiable from the sequence), we added the genes structure, and emphasized that six accessions with a loss of enzyme activity had an unidentified lesion due to sequence quality for this locus. We added an additional sup table presenting the frequency calculations of the mutated accessions.

Reviewer #3:The manuscript is based on an impressive data set of (seed) glucosinolate profiles across Europe and beyond, with a focus on methionine-derived glucosinolates. However, the manuscript reads in many aspects like a 'best of' of previous research, with limited net gain in insight. In addition, there are substantial weaknesses (in particular relating to GS-Elong) that prevent me from recommending this manuscript for publication in eLife its current state.1. The representation of the MAM reaction is misleading and incorrect (lines 101ff). MAMs catalyze condensation of 2-oxoacids with acetylCoA. Two further reaction steps, catalyzed by an isomerase and a dehydrogenase, are necessary to complete chain extension. This extension consists formally in the addition of a methylene group per reaction cycle. With MAM2, the net result is an extension by one (not two!) methylene group, with MAM1 it is mostly two; note that methionine has already two methylene groups in the side chain! I suggest replacing the reference Abrahams et al., 2020, with Graser et al., 2000 (Archives of Biochemistry and Biophysics, 378, 411-419). Furthermore, Kroymann et al., 2001 (Plant Phys. 127, 1077-1088), Benderoth et al., 2006 and Textor et al., 2007 (Plant Phys. 144, 60-71) should be added as original references for the MAM1, 2 and 3 reactions.

We apologize for the error and have rewritten this section, corrected the mistakes, and updated the citations.

2. The use of MAM3 as a proxy for inferring the configuration of the GS-Elong locus is highly debatable. Even though recombination appears to be suppressed at and around GS-Elong, an additional gene (or additional genes) upstream of GS-Elong should be used to construct (a) gene tree(s) and compare with the MAM3 gene tree for congruency. In any case, without sequencing of the MAM genes at the GS-Elong locus, it is not possible to conclude that 'local' configurations of MAM1/MAM2 have not arisen in Iberia (or elsewhere).

To address this concern, we have generated an additional phylogenetic tree using the sequences of MYB37, the gene on the opposite end of the GS-Elong locus (At5g23000), from all the 797 accessions. To test for within locus recombination that may be confusing us, we compared this tree to the MAM3. The vast majority of accessions showed the same major clade memberships in both the MAM3 and MYB37 trees arguing against a high level of within locus recombination. We also reanalyzed our analysis using just the 637 accessions with perfect agreement in the two trees and these showed the same message. Thus, our overarching patterns are not confounded by recombination shuffling this locus.

We realized that we had been inaccurate when discussing a local Iberian haplotype as we’d meant to mean a single haplotype explaining all C3/C4 diversity Iberia. The reviewers comment led us to search for possible true local haplotypes by using the few accessions that have an incongruency in the MYB37/MAM3/C3/C4 data. This identified a collection of 6 accessions in a very narrow region of Iberia that appear to have developed a new local haplotype that requires further sequencing. We added a supp. Figure that describes the detection of these accessions and added this comment to the manuscript. Additionally, we have obtained new genomic sequences of the MAM locus for 11 accessions that resample the existing haplotypes and sample new haplotypes from the clades previously not sampled. These sequences support the classification of the accessions to the clades based on the MAM3 sequences, and the use of the MAM3 gene as a proxy to the MAM1/MAM2 haplotypes. In combination this suggests that while there is at least one local haplotype in Iberia, the vast majority of accessions in Iberia don’t have unique events, and the presence of these few accessions will not change the overall message.

3. The A. lyrata ortholog of Arabidopsis MAM3 is MAMb (not MAMc). The MAM3 gene tree does not provide any statistical support for the branching order. Therefore, it remains unclear whether the presented tree reflects the true branching order and whether the 'most basal' clade in the tree is indeed the most basal. Note also that the Cal-0 gene at the MAM2 position is actually mostly MAM2, except for a converted stretch towards the 3' end of the gene; see: Figure 2 in Kroymann et al., 2003. Hence, the most parsimonious interpretation of available data is that the ancestral state of GS-Elong in Arabidopsis is indeed: MAM2, MAM1 and MAM3. Note also that the Arabidopsis genes in Abrahams et al., 2020, are from TAIR 10, i.e. from Col-0 (see Table 1 in Abrahams et al., 2020). Because Col-0 does not have MAM2, this gene is not included in the gene trees presented by Abrahams et al., 2020.

The MAM3 tree was indeed rooted by *A. lyrata* MAMb, and not MAMc as was written in the manuscript. This was a typo that was corrected. As for the gene tree presented in Abrahams et al., – we teamed up with them to recreate the tree, this time including MAM2 (supp. Figure x).

For the MAM3 tree, we have worked in several ways to determine the proper support for this tree and the claims. The tree with all accessions has some sequences that are not maximal quality which influences the bootstrapping but not the clade assignment. To improve our precision, we created a tree using 637 accessions that have the highest quality accessions for MAM3 (supp. Figure 10). The accessions in the new tree classified to the same clades as in the original MAM3 tree. The only difference between the trees (original tree and the cleaner tree) was the order of the clades. We then created multiple smaller trees, each contain a different combination of 200300 accessions. In each of these trees the accessions classified to the same clades as the original MAM3 tree, with bootstrap support. This agrees with the reviewer’s assessment that the evolutionary relationships between the clades are not resolvable but that the clades are decently defined. As we don’t have a strong support to the orders of the clades, we soften our tone in the manuscript in the part talking about the clades order and the ancestral state of the locus. We have discussed the most functionally parsimonious model of the ancestral state being MAM2/1/3 but also noted that this model requires a MAM1/2 duplication and diversification following the split from *A. lyrata* and ensuing structural variation to create the haplotypes.

4. Despite the use of an impressive number of accessions, sufficiently dense sampling appears to be only available for the Iberian peninsula, the southern coast of Sweden and the south-western coast of Italy. In this regard, the authors overstate their findings (e.g., lines 219ff).

We added this clarification to several locations along the text (results and discussion) and soften our claims.

5. While the link between glucosinolates and biotic stress is well documented, evidence for a major role of abiotic factors in shaping glucosinolate profiles is less well substantiated. Now, the authors seem to find that certain chemotypes are favored by high precipitation in 'the North' but that the same chemotypes are favored by low precipitation in 'the South'. This is interesting, but what could explain this unexpected pattern? Any idea? Furthermore, with additional sampling on the eastern coast of Italy and the eastern coast of the Adriatic, would you expect to obtain similar patterns?

We think what is happening is that the abiotic factors are capturing the biotic environment but there are no universal databases describing the biotic environment. We have added the following to try and describe this “The above analysis is extensively using Abiotic factors because of their availability while Aliphatic GSLs have mainly been linked biotic interactions. Our best hypothesis is that the abiotic factors used in this model are capturing variation in the biotic environments. However, an Aliphatic GSL was recently mechanistically linked to drought resistance in Arabidopsis (Salehin et al., 2019). This suggests a need to capture the biotic as well as the abiotic components of any given environment to distinguish what may be driving these patterns.”

6. Please, make sure that the citation style is uniform. Please also make sure that the tense is uniform throughout the manuscript. Please, simplify the introduction in a way that it is accessible by non-specialists. Please improve Figure 5D – this is a mess.

We have worked to correct tense and citation style. We re-organized the whole figure to make it cleaner and clearer. We moved part D to the supplementary so part C (the map) will have more space. Hopefully it helped to make things less messy. We have worked on the introduction and have run it past five individuals who are not glucosinolate specialists and they have helped greatly.

[Editors’ note: what follows is the authors’ response to the second round of review.]

The reviewers have discussed their reviews with one another, and the Reviewing Editor has drafted this to help you prepare a revised submission.Essential Revisions:While we find the overall story to be insightful and advancing the field, we do think that some of the analyses still need to be redone, while the eco-evolutionary framework needs to be better defined.1. We have serious concerns regarding the environmental association analyses. Since there is a general correlation between genetic/genomic/phenotypic diversity and environmental factors due to Arabidopsis demographic history, the overall genetic structure needs to be included in the statistical models testing the associations between traits and environmental factors. You should produce tables similar to that of Figure 4—source data 1 for the chemotypes of Figure 4 and derive statistical significances from these tables, in order to provide convincing evidence of the differential climatic patterns in both European regions.

We apologize as we had run all the models with and without genomic group in them and did not include this table. We have rerun the models to include the genomic group to account for population structure. As suggested, for Figure 4 (now Table 1) we created a table in the same manner as the one in supp figure 8 that now accounts for structure. All the p-values in the analyses are derived from models that include the genomic group. We rewrote this section of the results, discussion and the methods accordingly.

2. The conceptual framework of repeated evolution that differentiates between parallel and convergent evolution was traditionally developed for interspecific evolution, not for intraspecific evolution where it refers to parallel evolution by multiple independent mutations in the same gene. The differentiation between parallel and convergent evolution for intraspecific evolution is not so straightforward. Therefore we want you to provide a more precise definition of intraspecific convergent and parallel chemotype evolution and outline the role of different or the same genes from a tandem repeat cluster, as well as the role of epistasis between duplicated genes, therein. To this end, the LD analysis between MAM genes and flowering genes should also be removed from the argumentation, because this analysis clearly shows that LD between these genes is not larger than average LD between random pairs of genes. Therefore, the conclusion drawn from this LD analysis is not supported.

We rewrote the sections concerning parallel/convergent evolution throughout the manuscript (more details are below) and added an introductory figure that will hopefully facilitate a better conveyance of how we are attempting to incorporate these terms in this work. The figure and the section concerning the trans-LD analysis were removed from the manuscript according to the reviewers’ suggestion.

We are optimistic that regardless of the outcome of the corrected environmental analysis (point 1), this study represents a valuable contribution, especially if the revised framework for intraspecific repeated evolution – helping to conceptually unite studies of population and functional genetics in an apt system – can be articulated and illustrated more clearly in the revised analysis (point 2).Reviewer #1 (Recommendations for the authors):In this new version of the Manuscript by Katz et al., authors address numerous important questions raised in the previous version, and rewrite many of the paragraphs that were confusing or incorrect. In this version authors provide better support for some of their conclusions in a more comprehensive manuscript. However, this Reviewer still has serious concerns regarding the environmental association analyses carried out in this study to support their conclusions. In addition, for clarity and comprehension I also suggest addressing an important conceptual issue, as well as the numerous editorial points described below.1A. As explained in previous Reviewer 2 report to Authors, in *A. thaliana* there is a general correlation between genetic/genomic/phenotypic diversity and environmental factors due to Arabidopsis demographic history. For this reason, the overall genetic structure needs to be included in the statistical models testing the associations between traits and environmental factors. Authors now provide more detailed description of the statistical tests applied for such associations. However, as stated in Materials and methods, the statistical models used to build Figure 4 and Supplemental Figure 8B do not include the genomic groups. Therefore, significant associations described in these Figures and throughout the manuscript are mainly reflecting the association of genomic groups (demographic history) with climate. These two figures are the main support for the differential environmental associations in Northern and Southern Europe, but they just show that different genomic groups are present in each geographic area. It seems that such differential association between climate and chemotypes might be true, as shown by the table of Supplementary Figure 8A, where authors include the genomic groups to test the differential association specifically for C3 and C4 status. However, in this table the variable "Precipitation for the driest moth" appears as non-significantly associated. In spite of this, authors choose this variable in Supplementary Figure 8B (as well as in Figure 4A) to illustrate that this climatic variable is associated when applying a simple t-test to compare both groups. This test does not support such environmental association because it is not controlling for the genomic group (demographic history). By contrast, the table of Supplementary Figure 8 shows a stronger differential association in Northern and Southern Europe for minimum temperature, which is associated only in the south but not in the north. Tables similar to that of Supplementary Figure 8A should be provided for the chemotypes of Figure 4, and statistical significances should be derived from them, in order to provide convincing evidence of their differential climatic patterns in both European regions.

We rerun the models, and now all the models include the genomic group. As suggested, for Figure 4 (now Table 1) we created a table in the same manner as the one in supp figure 8. All the p-values in the analyses are derived from models that include the genomic group. We rewrote the corresponding sections accordingly.

1B. Related with the environmental associations, authors include random forest analyses. However, in the way these analyses are presented, they are hard to interpret because variables with the lowest mean decrease Gini (the probability of the variable of being wrong according to Authors) mostly have the lowest mean decrease accuracy (suitability of the variable as predictor).

We had intended the random forest to convey the differential ranks of the different genetic and environmental parameters across geography. We agree that this is not well shown in the current representation and we removed this analysis from the manuscript.

2. A major goal of this manuscript is the distinction between convergent and parallel evolutions in GSL chemotypes, which are described as two different patterns of repeated evolution. However, these concepts have been classically developed to characterize interspecific evolution, and their translation into the intraspecific variation is not straight forward, thus requiring a precise definition of both alternatives. Authors define parallel and convergent evolution as traits derived from either the same or different "genetic backgrounds". In this definition, the concept of genetic background is very vague and it is unclear what is meant. Authors apply these two concepts to the particular case of repeated evolution of GSL chemotypes evolved specifically from two tandem duplicated genes (MAM and AOP loci). Given the epistasis between these pairs of duplicated genes, the same trait can be derived from the same or different gene within. Then, it seems to this Reviewer that Authors mean by parallel and convergent evolution, the evolution occurring from the same or different gene of the cluster, respectively. In my opinion, this dissection of repeated evolution for the specific case of duplicated genes with epistasis is a major contribution of this study, but it is hard to understand. The manuscript and the audience would benefit by making these concepts clearer. I suggest that Authors add this in discussion, (e.g. paragraph of lines 550-554), explaining the role of different or the same gene from tandem repeat cluster, as well as the role of epistasis between duplicated genes for the distinction of their concepts of parallel and convergent evolution.

We rewrote this section to emphasize the above points. For the discussion we split this into two paragraphs, one on epistasis and one on convergent/parallel to better convey these concepts. We have also reworked the initial paragraph within the introduction to try and translate the interspecific concept to the intraspecific analogy we are contemplating. Specifically that genetic background is a specific haplotype and that if parallel evolution would be a new chemotype arising by independent mutations in a single haplotype to give the new chemotype multiple times. Convergent evolution would be where a new chemotype arose by independent mutations in different haplotypes both giving rise to a single new chemotype. To make sure that our idea is clear we added an introductory figure (Figure 1) that presents how we are trying to incorporate these terms in this work.